# Pre-symptomatic Caspase-1 inhibitor delays cognitive decline in a mouse model of Alzheimer disease and aging

Joseph Flores [1,2], Anastasia Noël [1,2], Bénédicte Foveau[1], Olivier Beauchet[1,3,4] & Andréa C. LeBlanc [1,2,5✉]

Early therapeutic interventions are essential to prevent Alzheimer Disease (AD). The association of several inflammation-related genetic markers with AD and the early activation of pro-inflammatory pathways in AD suggest inflammation as a plausible therapeutic target. Inflammatory Caspase-1 has a significant impact on AD-like pathophysiology and Caspase-1 inhibitor, VX-765, reverses cognitive deficits in AD mouse models. Here, a one-month pre-symptomatic treatment of Swedish/Indiana mutant amyloid precursor protein (APP$^{Sw/Ind}$) J20 and wild-type mice with VX-765 delays both APP$^{Sw/Ind}$- and age-induced episodic and spatial memory deficits. VX-765 delays inflammation without considerably affecting soluble and aggregated amyloid beta peptide (Aβ) levels. Episodic memory scores correlate negatively with microglial activation. These results suggest that Caspase-1-mediated inflammation occurs early in the disease and raise hope that VX-765, a previously Food and Drug Administration-approved drug for human CNS clinical trials, may be a useful drug to prevent the onset of cognitive deficits and brain inflammation in AD.

---

[1] Bloomfield Center for Research in Aging, Lady Davis Institute for Medical Research, Jewish General Hospital, Montreal, Quebec, Canada. [2] Department of Neurology and Neurosurgery, McGill University, Montreal, Quebec, Canada. [3] Department of Medicine, Division of Geriatric Medicine, Sir Mortimer B. Davis - Jewish General Hospital, Montreal, Quebec, Canada. [4] Lee Kong Chian School of Medicine, Nanyang Technological University, Singapore, Singapore. [5] Department of Anatomy and Cell Biology, McGill University, Montreal, Quebec, Canada. ✉email: andrea.leblanc@mcgill.ca

There is an urgent need to assess alternative early therapeutic targets against Alzheimer disease (AD). Aβ, Tau protein, neurotransmitters, and metabolic pathways represent the four main therapeutic targets in AD[1]. Cholinesterase inhibitors and the memantine N-methyl-D-aspartate acid (NMDA) antagonist are standard AD-specific symptomatic drugs which have shown beneficial but modest clinical effects against cognitive decline in mild-to-moderate AD[2]. Disease modifying immunotherapies against Aβ, predominantly studied in clinical trials, have yet to demonstrate clinical efficacy[3]. Tau therapies are still in early developmental stages[4].

Aging is the most prevalent risk factor for AD and provides insight into disease etiology and progression. Inflammation is an altered physiological process during senescence and there are age-related changes in immune-related genes[5], region-specific immunophenotype[6], and microglia morphology[7], that can prime the brain for neurodegeneration and age-related cognitive decline. Genetic and experimental evidence also support a pathogenic role for inflammation in AD[8,9]. The common mediators between aging and AD are predominantly pro-inflammatory and involve the increased expression of various cytokines, including interleukin-1-beta converting enzyme (IL-1β), interleukin (IL)-6, IL-18 and tumour necrosis factor alpha (TNF-α). Dysregulation of the inflammatory system in aging and AD could impact brain function and facilitate cognitive impairment.

Early epidemiological studies have shown an association between long-term use of non-steroidal anti-inflammatory drugs (NSAIDs) and lower AD risk[10]. NSAIDs reduce inflammation, minimize Aβ accumulation and improve cognition in AD animal models[11], but have disappointedly not prevented or improved age-dependent cognitive loss or AD in clinical trials[12,13]. Different NSAIDs are being considered to improve treatment efficacy[14]. Recently, danger- or pathogen-associated molecular patterns (DAMPs and PAMPs) have been linked via inflammasome and Caspase-1 (Casp1) activation to the excess IL-1β found in AD brains[15–17]. Inhibition of the inflammasome Nod-like receptor protein 1 (Nlrp1)[18], Nlrp3[15], Casp1[15,19] or the inflammasome activator purinergic P2X7 receptor[20], reduces inflammation, improves synaptic pathology, reduces Aβ accumulation and reverses cognitive deficits in AD mouse models. Similarly, knockout or immunotherapy against the inflammasome adaptor protein, apoptosis-associated speck-like protein containing a caspase activated and recruitment domain (ASC), decreases Aβ deposition and improve cognitive behaviour in the amyloid precursor protein/presenilin 1 mutant (APP/PS1) AD mouse model[17].

The Nlrp1–Casp1-Casp6 pathway represents an early therapeutic target against AD[16]. This pathway links Nlrp1- and Casp1-mediated inflammation to Casp6-mediated neurodegeneration and AD pathologies including Aβ increased production, Tau cleavage, misfolded protein proteasomal clearance problems, cleavage of several cytoskeletal and synaptic proteins, and IL-1β production[16,21–24]. In human primary CNS neuron cultures, Nlrp1 inflammasome-mediated Casp1 activation activates neurodegeneration-related Casp6[16]. Transfection of human neurons with mutant or wild-type APP induces Casp6-dependent, but Aβ-independent, neuritic degeneration[25]. Active Casp6 in the hippocampal Cornu Ammonis 1 (CA1) is sufficient to induce age-dependent cognitive deficits in mice[26]. Theta-burst long-term potentiation cannot be initiated in mice acute brain slice hippocampal CA1 neurons expressing active Casp6[27]. Furthermore, injection of active Casp6 in wild-type mice CA1 neurons impairs synaptic transmission and induces neurodegeneration[28]. Active Casp6 and Nlrp1 are highly increased and co-localized in AD neurons. Active Casp6, which is abundantly present in AD neuritic plaques, pre- and mature neurofibrillary tangles, and neuropil threads[22,29], correlates negatively with episodic and semantic memory performance in aged human individuals[29,30].

Inflammasome and Casp6 inhibitors are not yet available for human clinical trials. However, Casp1 inhibitors have been developed to treat inflammatory diseases[31]. Casp1 knockout in APP/PS1[15] or APP$^{Sw/Ind}$ mutant J20[19] AD mouse models improve cognitive behaviour and synaptic function, and decrease Aβ levels. VX-765, a blood–brain barrier permeable and non-toxic Casp1 inhibitor, reverses spatial and episodic memory deficits, inflammation and synaptic loss, and prevents progressive Aβ accumulation when given at the onset of the J20 mouse symptoms[19]. VX-765 has also been shown to reduce inflammation by suppressing CD4 T-cell death and pyroptosis in HIV[32], reduce inflammasome-related gene expression, axonal injury, improve behaviour in an experimental allergic encephalitis multiple sclerosis mouse model[33], and protect against alpha-synuclein toxicity both in vitro[34] and in the proteolipid protein-synuclein mouse model of multiple systems atrophy[35]. VX-765 is an orally absorbed prodrug of the active metabolite, VRT-043198. The IC$_{50}$ is 3.68 and 9.91 nM[19] and in vitro half-life ($t_{1/2}$) is ~157 and 3 h for VX-765 and VRT-043198, respectively[36]. A 50 mg • kg$^{-1}$ dose of VX-765 reversed cognitive deficits in early symptomatic J20 mice reaching ~2–5 μM concentrations of VX-765 and VRT-043198 in the J20 brain after carotid infusion[19]. Here, we show that VX-765 prevents AD-related memory deficits and pathologies in the J20 AD mouse model.

## Results

**VX-765 pre-symptomatic treatment delays cognitive deficits.** To determine if sufficient VX-765 reaches the brain after intraperitoneal (IP) injections at 50 mg • kg$^{-1}$, VX-765 and VRT-043198 concentrations were measured by LC-MS/MS in the plasma, whole brain, and cerebrospinal fluid (CSF) at 0.25, 0.5, 1, 3, 6, 8 and 24 h after a single intraperitoneal injection (Fig. 1). VX-765 and VRT-043198 reached plasma concentrations 100- and 20,000-fold (Table 1 and Fig. 1a), brain concentrations 50- and 170-fold (Table 2 and Fig. 1b), and CSF concentrations 100- and 4000-fold (Table 3 and Fig. 1c) their respective in vitro Casp1 IC$_{50}$[19] within 0.25 h post-injection. VRT-043198 levels surpassed VX-765 concentrations, which indicates the rapid conversion of the prodrug, VX-765, into VRT-043198 in all three compartments. The $t_{1/2}$ half-life of VX-765 was 3.2 h in plasma (Table 1) but could not be accurately quantified in brain (Table 2) or in CSF (Table 3). The $t_{1/2}$ of VRT-043198 was 1 h in brain but could not be calculated in plasma or in CSF. Although brain VX-765 and VRT-043198 levels were short-lived, the LC-MS/MS 0.025 and 0.049 μM lower limit of quantification for VX-765 for VRT-043198, respectively, prevented the detection of the low nM concentrations inhibiting Casp1. Nevertheless, these results confirm VX-765's good clinical profile, particularly its stable plasma levels. Most importantly, the data confirms that VX-765 and VRT-043198 reach the brain at concentrations sufficient to inhibit Casp1 after a single IP injection of VX-765.

To determine whether pre-symptomatic inhibition of Casp1 can slow cognitive deficits, 2-month-old wild-type (WT) and APP$^{Sw/Ind}$ mutant J20 mice were treated for 1 month with 3 injections per week of 50 mg·kg$^{-1}$ VX-765. Mice were behaviourally assessed at 4 months of age, 4 weeks after their last injection, when they normally exhibit cognitive deficits and hyperactivity, and again at 6 and 8 months of age (4-, 12- and 20-week washout [WO]) (Fig. 2a). To ensure a complete analysis of the J20's behavioural progression, a smaller number of mice were also behaviourally assessed at 8- and 16-week WO. At each WO

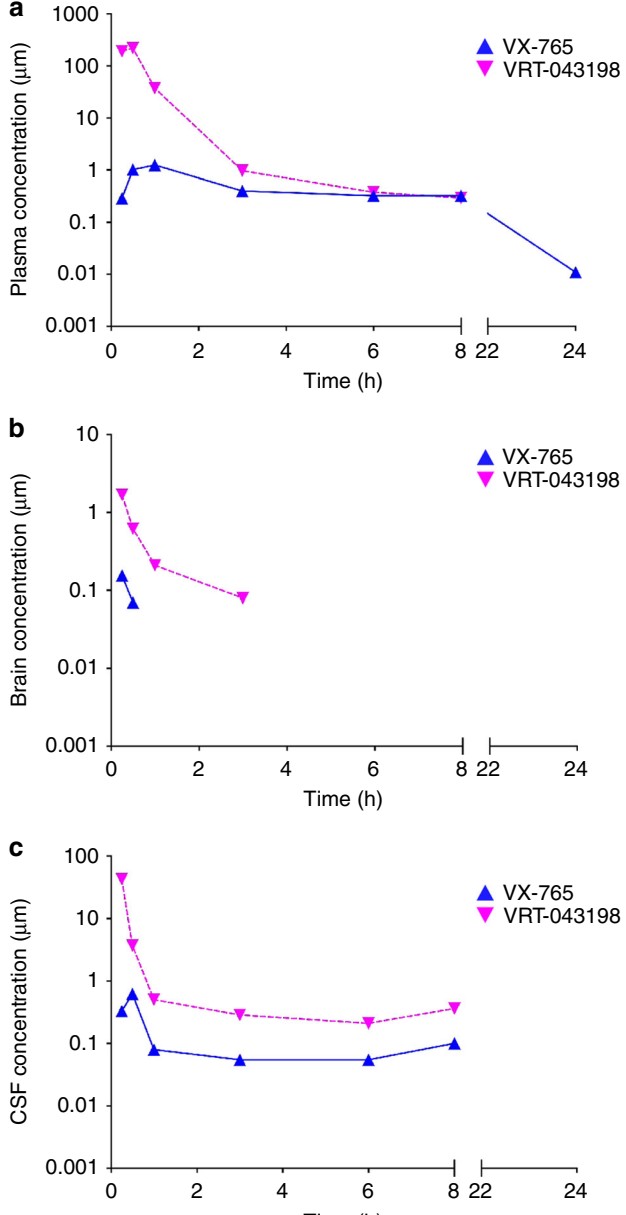

**Fig. 1 VX-765 and VRT-043198 are detected in mice after a single dose of VX-765. a–c** PK measures of VX-765 and active VRT-043198 metabolite in **a** plasma, **b** whole brain homogenates, and **c** CSF samples up to 24 h after a single IP injection of 50 mg·kg⁻¹ VX-765 ($n = 2$ mice per time point).

period, a subset of animals was euthanized for post-mortem analyses.

Mice were repeatedly tested between 4- and 20-week WO with novel object recognition (NOR) to assess episodic memory. A strong impairment was found in vehicle-treated J20 compared to vehicle-treated WT mice at 4-, 8-, 12-, 16- and 20-week WO (Fig. 2b and Supplementary Fig. 1a). VX-765 treatment protected J20 mice against NOR impairment at all WO periods. NOR performance in the groups of mice utilized for post-mortem analyses corroborated these findings (Supplementary Fig. 1b). Nevertheless, NOR performance declined between 4- and 20-week WO (Fig. 2b) in VX-765-treated J20. The percentage of NOR impaired mice increased consistently between 4- and 20-week WO in VX-765-treated J20 mice (Supplementary Fig. 1c).

Together, these results show that VX-765 delays the onset of cognitive impairment in J20 mice by at least 5 months, but there is a gradual loss of VX-765's effect over time, as expected from a drug washout.

In WT mice, NOR performance decreased in 8-month-old (20-week WO) compared to 4-month-old (4-week WO) vehicle-treated mice (Fig. 2b). In contrast, NOR performance increased in VX-765-treated 8-month-old (20-week WO) mice compared to same age vehicle-treated WT mice, and in VX-765-treated 6-month-old (12-week WO) compared to VX-765-treated 4-month-old WT mice. Consistently, the percentage of NOR impaired mice increased between 12- and 20-week WO in the vehicle-treated, but not VX-765-treated WT mice (Supplementary Fig. 1c). These results indicate VX-765 protection against age-dependent cognitive impairment in WT mice.

In mice tested repeatedly to 20-week WO (Fig. 2c and supplementary Fig. 1d) or analysed post-mortem at different WO times (Supplementary Fig. 1e), vehicle-treated J20 exhibited increased distance and quadrant entries in the open field compared to vehicle-treated WT mice at each WO period, consistent with previously reported hyperactivity[37] (Fig. 2c and Supplementary Fig. 1d–f). VX-765 treatment attenuated hyperactivity in J20 mice only at 4-week WO but had no effect on WT mice at any WO period. Even though WT mice gradually moved less with time (Fig. 2c) and were less active than J20 mice (Supplementary Fig. 1g), there were no differences in locomotor activity, time moving or thigmotaxis (Supplementary Fig 1h) between vehicle and VX-765 treatment. Therefore, cognitive differences assessed with NOR were not dependent on hyperactivity, motivation or anxiety.

Spatial learning and memory were examined with the Barnes maze at 12- and 20-week WO because spatial memory deficits appear at 6 months of age (12-week WO) in J20 mice. At 12-week WO, vehicle-treated J20 mice committed significantly more errors locating the target than vehicle- or VX-765-treated WT mice (Fig. 2d). VX-765 treatment prevented this learning deficit in J20 mice and reduced the number of errors committed to near WT levels by the end of the 4-day training phase. During training, primary latency to find the target did not differ between groups (Supplementary Fig. 1i). During the probe, carried out 24 h after the last training day to assess memory retention, primary latency did not differ between vehicle- or VX-765-treated J20 and vehicle-treated WT groups (Fig. 2e). Nevertheless, vehicle-treated J20 mice committed significantly more errors compared to WT and VX-765 treatment prevented this deficit. All groups, except the vehicle-treated J20 group, showed a preference towards the target area (Fig. 2f).

Similarly, vehicle-, but not VX-765-treated, J20 mice exhibited learning deficits at 20-week WO (Fig. 2g and Supplementary Fig. 1i). Vehicle-treated J20 mice were slower and committed more errors to reach the target compared to WT mice, and these deficits were prevented with VX-765 treatment (Fig. 2h, i).

Interestingly, VX-765-treated WT mice at 12-week WO had decreased primary latency and primary errors compared to vehicle-treated WT mice (Fig. 2e, h). This effect was attenuated at 20-week WO, although both vehicle- and VX-765-treated WT mice showed preference for the target area during the probe test (Fig. 2f, i).

Together, these results indicate that pre-treatment with VX-765 can significantly delay the onset of episodic and spatial memory deficits, but not hyperactivity, in J20 mice up to 20 weeks after treatment. In addition, VX-765 improves WT mice memory performance. Together, these preclinical results raise hope that pre-symptomatic treatment with VX-765 might prevent cognitive deficits in humans.

**Table 1 VX-765 and VRT-043198 are detected in plasma after a single IP dose of VX-765.**

| Time (h) | Mouse 1 VX-765 (µM) | Mouse 2 VX-765 (µM) | Mouse 1 VRT-043198 (µM) | Mouse 2 VRT-043198 (µM) | Mean VX-765 (µM)[a] | Mean VRT-043198 (µM)[b] |
|---|---|---|---|---|---|---|
| 0.25 | 0.31 | 0.27 | 201.8 | 180.4 | 0.29 | 191.1 |
| 0.50 | 1.54 | 0.49 | 338.1 | 104.1 | 1.01 | 221.1 |
| 1 | 0.87 | 1.63 | 24.4 | 51.0 | 1.25 | 37.7 |
| 3 | 0.45 | 0.35 | 1.16 | 0.78 | 0.40 | 0.97 |
| 6 | 0.43 | 0.22 | 0.46 | 0.30 | 0.33 | 0.38 |
| 8 | 0.2 | 0.45 | 0.35 | 0.24 | 0.32 | 0.30 |
| 24 | 0.008 | 0.014 | NQ | NQ | 0.01 | – |
| $C_{max}$ (µM)[c] | 1.54 | 1.63 | 338.1 | 180.4 | 1.59 | 259.3 |
| $T_{max}$ (h)[d] | 0.50 | 1.00 | 0.50 | 0.25 | 0.75 | 0.40 |
| Half-life ($t_{1/2}$) (h)[e] | 3.22 | 3.17 | NR | NR | 3.2 | – |
| $MRT_{inf}$ (h)[f] | 4.80 | 5.64 | 0.67 | 0.78 | 5.22 | 0.70 |

NQ: not quantifiable. No peak of below lower limit of quantification: 0.05 µM for VX-765 and 0.098 µM for VRT-043198.
NR: not reported. Only two points used for terminal slope calculation or the span of the terminal phase is too short to allow for accurate estimation of half-life.
Individual data points showing VX-765 and VRT-043198 concentrations up to 24 h post-injection ($n = 2$ per time point). (a) VX-765 and (b) VRT-043198 mean concentrations at each time point. (c) Maximum concentration achieved ($C_{max}$), (d) time to reach $C_{max}$ ($T_{max}$), (e) half-life ($t_{1/2}$) and (f) mean residence time (MRT) in plasma for VX-765 and VRT-043198.

**Table 2 VX-765 and VRT-043198 are detected in the brain after a single IP dose of VX-765.**

| Time (h) | Mouse 1 VX-765 (µM) | Mouse 2 VX-765 (µM) | Mouse 1 VRT-043198 (µM) | Mouse 2 VRT-043198 (µM) | Mean VX-765 (µM)[a] | Mean VRT-043198 (µM)[b] |
|---|---|---|---|---|---|---|
| 0.25 | 0.24 | 0.07 | 2.54 | 0.85 | 0.16 | 1.70 |
| 0.50 | 0.05 | 0.09 | 0.59 | 0.65 | 0.07 | 0.62 |
| 1 | NQ | NQ | 0.21 | 0.27 | – | 0.24 |
| 3 | NQ | NQ | 0.08 | NQ | – | 0.08 |
| 6 | NQ | NQ | NQ | NQ | – | – |
| 8 | NQ | NQ | NQ | NQ | – | – |
| 24 | NQ | NQ | NQ | NQ | – | – |
| $C_{max}$ (µM)[c] | 0.24 | 0.09 | 2.54 | 0.85 | 0.16 | 1.70 |
| $T_{max}$ (h)[d] | 0.25 | 0.50 | 0.25 | 0.25 | 0.38 | 0.25 |
| Half-life ($t_{1/2}$) (h)[e] | NC | NC | 0.99 | NR | – | 0.99 |
| $MRT_{inf}$ (h)[f] | NC | NC | 0.98 | 0.76 | – | 0.87 |

NQ: not quantifiable. No peak of below lower limit of quantification: 0.025 µM for VX-765 and 0.049 µM for VRT-043198.
NC: not calculated. Not enough data points.
NR: not reported. The span of the terminal phase is too short to allow for accurate estimation of half-life.
Individual data points showing VX-765 and VRT-043198 brain concentrations up to 24 h post-injection ($n = 2$ per time point). (a) VX-765 and (b) VRT-043198 mean concentrations at each time point.
(c) Maximum concentration achieved ($C_{max}$), (d) time to reach $C_{max}$ ($T_{max}$), (e) half-life ($t_{1/2}$) and (f) mean residence time (MRT) in the brain for VX-765 and VRT-043198.

**VX-765 pre-symptomatic treatment does not alter APP levels.** To determine if VX-765 affects APP expression and processing and could explain the delay in cognitive deficits, real-time PCR and western blots against full-length APP and APP C-terminal fragments (CTF) were performed. VX-765 did not alter human *APP* mRNA levels in the hippocampus at 4-, 12- or 20-week WO (Fig. 3a). Human APP protein levels in J20 mice at 4-week WO were variable and consisted of both mature and immature forms (Fig. 3b). By 20-week WO, variability between J20 mice was reduced and more mature than immature APP was present. Human APP protein levels, like mRNA levels, were not altered by VX-765 in the hippocampus at 4- or 20-week WO (Fig. 3c). Similar to the absence of human APP expression in WT mouse hippocampus (Fig. 3b), human APP was not detected in J20 peripheral tissues, confirming the specific expression of human APP in the J20 mouse brain (Supplementary Fig. 2a). Relative to endogenous mouse APP, APP protein levels, detected with the anti-APP C-terminal antibody that detects both human and mouse protein, were increased significantly in J20 mouse hippocampus confirming APP transgene overexpression (Supplementary Fig. 2b, c). APP levels were not affected by VX-765. There were no differences in βCTF and αCTF levels between vehicle- and VX-765-treated J20 mice at 4- or 20-week WO (Supplementary Fig. 2d, e), indicating that VX-765 does not affect APP CTF. Together, these results indicate that alterations in the APP[Sw/Ind] transgene expression cannot account for VX-765's beneficial cognitive effects.

**VX-765 pre-symptomatic treatment delays glial activation.** Microglial activation has long been implicated in AD etiology and is an early pathological feature preceding Aβ deposition in J20 mice[38]. Microglial activation involves increased inflammatory cytokine production, activation-dependent morphological changes and increased cell numbers[39]. Iba1, a marker for activated microglia and inflammation[40], was quantitatively measured from the pyramidal cell layer to the stratum lacunosum moleculare (SLM) of the hippocampal CA1, and from the retrosplenial area to the S1 in the cortex (Supplementary Fig. 3a). Iba1-positive microglia increased in the hippocampus and cortex of vehicle-treated J20, compared to WT mice, at all WO periods (Fig. 4a and Supplementary Fig 3b). VX-765 treatment prevented this increase, normalizing Iba1-positive numbers to near WT levels, although numbers did not reach significance in the hippocampus at 8- and 20-week WO.

**Table 3 VX-765 and VRT-043198 are detected in the CSF after a single IP dose of VX-765.**

| Time (h) | Mouse 1 VX-765 (μM) | Mouse 2 VX-765 (μM) | Mouse 1 VRT-043198 (μM) | Mouse 2 VRT-043198 (μM) | Mean VX-765 (μM)[a] | Mean VRT-043198 (μM)[b] |
|---|---|---|---|---|---|---|
| 0.25 | 0.21* | 0.45 | 82.6* | 3.11 | 0.33 | 42.9 |
| 0.50 | 0.07 | 1.17 | 0.44 | 7.00 | 0.62 | 3.72 |
| 1 | 0.06 | 0.10 | 0.43 | 0.58 | 0.08 | 0.50 |
| 3 | 0.07* | 0.04* | 0.42* | 0.15* | 0.06 | 0.28 |
| 6 | 0.05* | 0.06* | NQ* | 0.21* | 0.06 | 0.21 |
| 8 | 0.07* | 0.13* | 0.13* | 0.60* | 0.10 | 0.36 |
| 24 | NQ | NQ* | NQ | NQ* | — | — |
| $C_{max}$ (μM)[c] | 0.21 | 1.17 | 82.6 | 7.00 | 0.69 | 44.8 |
| $T_{max}$ (h)[d] | 0.25 | 0.50 | 0.25 | 0.5 | 0.38 | 0.38 |
| Half-life ($t_{1/2}$) (h)[e] | NC | NC | NR | NC | — | — |
| $MRT_{inf}$ (h)[f] | NC | NC | 0.81 | NC | — | 0.81 |

NQ: not quantifiable. No peak of below lower limit of quantification: 0.006 μM for VX-765 and 0.068 μM for VRT-043198.
NC: not calculated due to shape of PK curve.
NR: not reported. The span of the terminal phase is too short to allow for accurate estimation of half-life.
*Some blood contamination during CSF collection.
Individual data points showing VX-765 and VRT-043198 concentrations up to 24 h post-injection ($n = 2$ per time point). (a) VX-765 and (b) VRT-043198 mean concentrations at each time point. (c) Maximum concentration achieved ($C_{max}$), (d) time to reach $C_{max}$ ($T_{max}$), (e) half-life ($t_{1/2}$) and (f) mean residence time (MRT) in the CSF for VX-765 and VRT-043198.

Microglia were further quantified morphologically and four different profiles were identified based on ramified resting, reactive or amoeboid-activated and phagocytic-like morphologies (Supplementary Fig. 3c)[41]. Vehicle-treated J20 mice contained a lower percentage of resting type I and higher amount of type II activated microglia compared to WT mice in the hippocampus and cortex across all WO periods (Fig. 4b). VX-765 treatment reverted the profile in J20 mice back to a near WT profile at 4-week WO, but it slowly shifted from a resting to an activated state morphologically similar to vehicle-treated J20 mice from 12-week WO onwards. Type III, amoeboid-appearing, microglia were low at 4- and 12-week WO but significantly higher in vehicle- and VX-765-treated J20 mice at 20-week WO. Type IV microglia, which were rarely seen in WT and vehicle-treated J20 mice, were elevated in the hippocampus of VX-765-treated J20 mice at 20-week WO. To determine whether type IV microglia constitute phagocytic microglia, the type IV microglial profile observed at 20-week WO was further characterized using CD68 to identify phagocytic microglia and macrophages. CD68 positive staining was sparse in the hippocampus and cortex of J20 mice. Only a few CD68 positive cells resembling type IV microglia were identified, while the majority of cells that were CD68 positive resembled perivascular macrophages (Supplementary Fig. 4a). CD68 was seen only in VX-765-treated J20 mice: four out of the five VX-765-treated J20 mice showed CD68 positive staining whereas no staining was found in any vehicle-treated J20 (Supplementary Fig, 4b). This suggests that despite seeing increased Iba1-positive morphologically-identified phagocytic microglia after VX-765 treatment, the lack of CD68 staining makes it unlikely that microglial phagocytosis plays a direct role in VX-765's mechanism of action. Nevertheless, overall these results suggest that VX-765 can reduce microglial activation up to 4 months in the hippocampus and 5 months in the cortex of J20 mice and exert a long-term change on the morphology of microglia.

Hippocampal glial fibrillary acidic protein (GFAP) levels (Fig. 4c), measured by western blotting, increased in vehicle-treated J20 compared to WT mice at 4- and 20-week WO but were normalized with VX-765 treatment only at 20-week WO (Fig. 4d). Comparatively, immunopositive GFAP astrogliosis (Supplementary Fig. 5a) increased in vehicle-treated J20 at all WO periods in the hippocampus, and at 4- and 12-week WO in the cortex (Fig. 4e and Supplementary Fig. 5b). VX-765 treatment prevented astrogliosis until 8- and 12-week WO in the

hippocampus and cortex, respectively. The discrepancy between the western blot and immunostaining data at 4- and 20-week WO is possibly due to differences in the regions of interest analysed, or the dilution of GFAP with other cell proteins during extraction. Overall, these results suggest that VX-765 pre-treatment delays GFAP astrogliosis for 3 months in J20 mice.

**VX-765 pre-treatment normalizes IL-1β levels.** Activated Casp1 cleaves the inflammatory cytokines IL-1β and IL-18 into their active states, leading to IL-1β-mediated inflammation through microglial activation and astrogliosis. IL-1β and IL-18 were assessed to determine whether Casp1 is directly implicated in inflammation in the J20 brain. There were no differences in hippocampal *IL-1β* and *IL-18* mRNA levels between groups across any WO period, except for increased Il-1β mRNA in the J20 hippocampus that was normalized with VX-765 treatment at 16-week WO (Fig. 5a and Supplementary Fig. 6a). Pro-, active (Δ) and total (pro + active) IL-1β protein levels were measured by western blot in the hippocampus and cortex at 4- and 20-week WO (Fig. 5b). While hippocampal pro IL-1β levels remained unchanged (Fig. 5c, f), active and total IL-1β levels seemed higher in vehicle-treated J20 versus WT mice at both 4- and 20-week WO (Fig. 5d, e, g, h). This trend of increased active and total hippocampal IL-1β levels was not observed with VX-765 treatment. Active IL-1β levels were significantly higher in vehicle-treated J20 cortex but were normalized with VX-765 treatment at 20-week WO (Fig. 5g).

Total protein IL-1β levels did not differ between treatment groups in the hippocampus at 4-, 12-, or 20-week WO when measured by ELISA (Fig. 5i and Supplementary Fig. 6b). However, hippocampal IL-1β levels seemed to increase in vehicle-treated J20 versus WT mice at 12- and 20-week WO. This increase was not observed after VX-765 treatment. Cortical IL-1β levels did not differ between groups at any WO. The low constitutive expression of IL-1β[42,43] or its upstream activators in younger mice, and the delay between treatment cessation and IL-1β sampling could explain the absence of significant difference in IL-1β in these tissues.

Quantitative PCR was also performed to determine whether inflammasome-related proteins or AD-associated caspases were upregulated in the J20 hippocampus at the mRNA level. *Nlrp1*, *Nlrp3* and *Asc* hippocampal mRNA levels did not differ between the three treatment groups at any of the WO periods, although

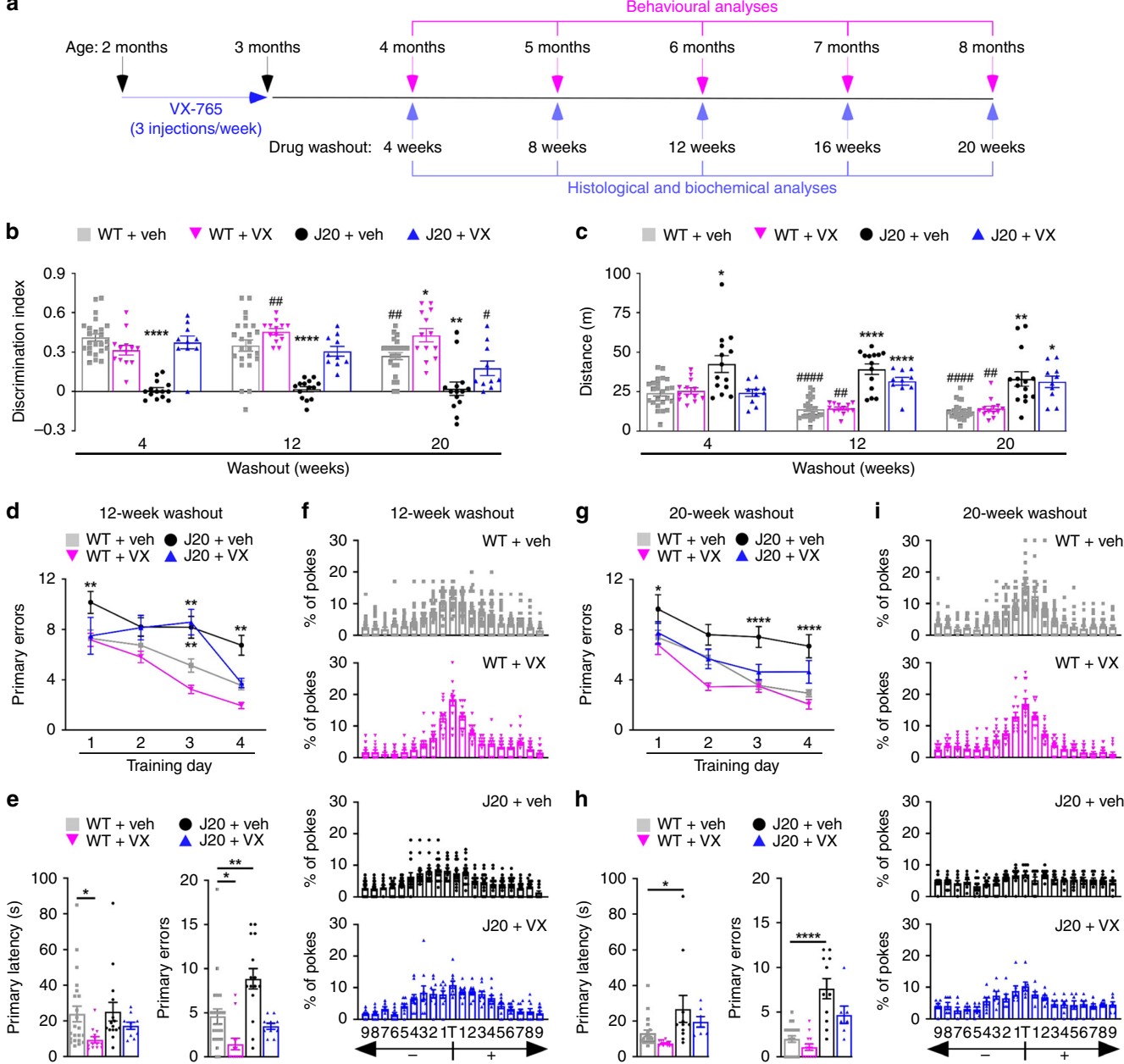

**Fig. 2 Pre-symptomatic VX-765 treatment delays cognitive deficits in J20 mice. a** Schematic diagram showing treatment and behavioural assessment. **b** NOR discrimination index of repeatedly tested vehicle-treated WT (□, $n = 24$), VX-765-treated WT (▽, $n = 13$), vehicle-treated J20 (●, $n = 14$) and VX-765-treated J20 (△, $n = 10$) mice at 4-, 12- and 20-week WO [treatment $F_{(3,57)} = 49.53$, $p = 1.0 \times 10^{-15}$; treatment × WO interaction $F_{(6,114)} = 3.576$, $p = 0.0028$, repeated-measures two-way ANOVA, Bonferroni's post hoc compared to same treatment group at 4-week WO (#) or Dunnett's post hoc comparing different treatment groups to WT + vehicle within same WO (*)]. **c** Open-field distance travelled of vehicle-treated WT ($n = 24$), VX-765-treated WT ($n = 13$), vehicle-treated J20 ($n = 14$) and VX-765-treated J20 ($n = 10$) mice [treatment $F_{(3,57)} = 33.38$, $p = 1.38 \times 10^{-12}$; WO $F_{(2,110)} = 6.856$, $p = 0.0018$; treatment × WO interaction $F_{(6,114)} = 3.801$, $p = 0.0017$, repeated-measures two-way ANOVA, Bonferroni's post hoc compared to same treatment group at 4-week WO (#) or Dunnett's post hoc comparing different treatment groups to WT + vehicle within same WO (*)]. **d–f** Barnes maze analyses of vehicle-treated WT ($n = 22$), VX-765-treated WT ($n = 13$), vehicle-treated J20 ($n = 15$) and VX-765-treated J20 ($n = 10$) mice at 12-week WO. **d** Learning acquisition primary errors [treatment $F_{(3,219)} = 20.58$, $p = 8.74 \times 10^{-12}$; training day $F_{(3,219)} = 22.94$, $p = 5.94 \times 10^{-13}$, two-way ANOVA, Dunnett's post hoc compared to WT + vehicle], **e** probe primary latency and primary errors [primary errors $F_{(3,56)} = 11.01$, $p = 8.76 \times 10^{-6}$, ANOVA, Dunnett's post hoc compared to WT + vehicle], and **f** percentage of pokes for each hole during probe, where T indicates target hole. **g–i** Barnes maze analyses of vehicle-treated WT ($n = 20$), VX-765-treated WT ($n = 13$), vehicle-treated J20 ($n = 11$), VX-765-treated J20 ($n = 8$)] mice re-tested at 20-week WO. **g** Learning acquisition primary errors [treatment $F_{(3,198)} = 24.05$, $p = 2.57 \times 10^{-13}$; training day $F_{(3,198)} = 23.69$, $p = 3.82 \times 10^{-13}$, two-way ANOVA, Dunnett's post hoc compared to WT + vehicle]. **h** Probe primary latency and primary errors [primary latency $F_{(3,47)} = 4.850$, $p = 0.0051$; primary errors $F_{(3,47)} = 19.75$, $p = 2.0 \times 10^{-8}$, ANOVA, Dunnett's post hoc compared to WT + vehicle], and **i** percentage of pokes during the probe. Data represents mean and s.e.m. Each symbol represents one mouse. *$p < 0.05$, **$p < 0.01$, ***$p < 0.001$, ****$p < 0.0001$ for all panels; #$p < 0.05$, ##$p < 0.01$, ####$p < 0.0001$ for panels (**b**) and (**c**).

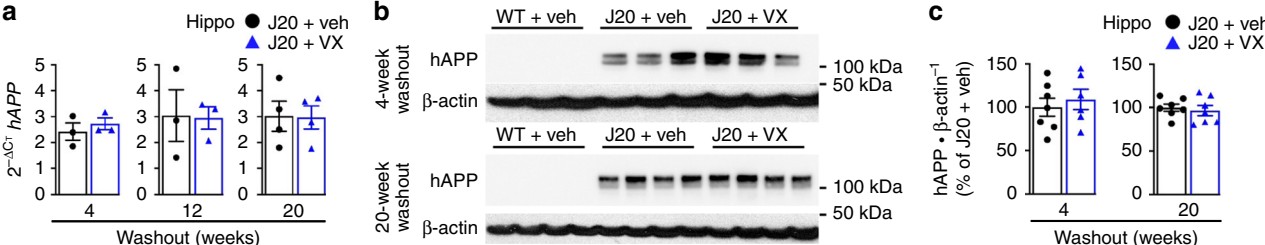

**Fig. 3 Pre-symptomatic VX-765 treatment does not alter APP expression in J20 mice. a** hAPP mRNA levels in the hippocampus at 4-week ($n = 6$ mice per group), 12-week ($n = 3$ mice per group) and 20-week ($n = 4$ mice per group) WO. **b** hAPP western blot and **c** quantification in the hippocampus at 4-week ($n = 7$ J20 + veh, 6 J20 + VX) and 20-week ($n = 7$ mice per group) WO using the human specific 6E10 antibody. Data in (**a**) and (**c**) represents mean and s.e.m.

there was a trend towards increased *Nlrp1* mRNA levels in vehicle-, but not VX-765-treated, J20 mice at 16- and 20-week WO (Supplementary Fig. 6c). Furthermore, no differences were observed in inflammation-related *Casp1*, *Casp6*, *Casp11* and apoptosis-related *Casp3* hippocampal mRNA levels (Fig. 5j and Supplementary Fig. 6d, e).

A western blot to detect full-length and processed active p33 and p20 Casp1 subunits was performed with proteins from vehicle-treated WT and J20 and VX-765-treated J20 cortex at 20-week WO (Supplementary Fig. 6f). Full-length Casp1 was detected in both WT and J20 mice at equivalent levels (Supplementary Fig. 6g). VX-765 treatment reduced full-length Casp1 levels slightly compared to vehicle-treated J20 mice, but this did not reach statistical significance. Cleaved-Casp1 active subunits were not detected in mouse cortical proteins despite being easily detected in the positive control tumour colon proteins and in lipopolysaccharide (LPS)/nigericin-treated J774A.1 microglial cell culture medium. Our inability to detect active Casp1 subunits, which contrasts with results obtained previously in the 16-month-old APP/PS1 mouse brains[15], likely reflects their low levels in 8-month-old J20 mouse brain or their rapid degradation after being secreted.

**VX-765 pre-symptomatic treatment does not alter Aβ levels.** Aβ plaques, a pathological hallmark of AD, begin to develop in the dentate gyrus and cortex of APP-overexpressing J20 mice between 5 and 7 months of age, with widespread deposition by 8–10 months of age[44]. In situ, Aβ deposition was concentrated in the SLM and dentate gyrus of the hippocampus and the cortical retrosplenial area in J20 mice and appeared unchanged after VX-765 treatment by 20-week WO (Fig. 6a). Aβ gradually accumulated with age and was more elevated in the hippocampus than cortex in J20 mice (Supplementary Fig. 7a). VX-765 significantly decreased Aβ staining density only in the cortex at 12-week WO, although Aβ staining seemed lower in the hippocampus in some WO periods, and from 4- to 12-week WO in the cortex (Fig. 6b and Supplementary Fig. 7b).

ELISA-measured RIPA-soluble total Aβ levels were higher in the hippocampus than the cortex (Fig. 6c and Supplementary Fig. 8a). There were no significant differences in RIPA-soluble Aβ38, Aβ40, Aβ42 or total Aβ levels between vehicle- and VX-765-treated J20 mice in the hippocampus or cortex across all WO periods (Supplementary Fig. 8b). However, RIPA-soluble Aβ42/total Aβ ratio was significantly higher in vehicle- compared to VX-765-treated hippocampus in J20 mice at 20-week WO (Fig. 6c).

Similar to Aβ immunohistological results, formic acid (FA)-soluble total Aβ levels increased after 12-week WO but did not differ between vehicle and VX-765 treatment in the hippocampus or cortex of J20 mice at any WO period (Fig. 6d and Supplementary Fig 8c). VX-765 treatment increased FA-soluble

hippocampal Aβ38 and Aβ40 levels at 20-week WO but did not affect hippocampal Aβ42 levels and Aβ42/total Aβ ratio, or any Aβ subtype in the cortex (Fig. 6d and Supplementary Fig. 8d). Together, these results suggest that pre-treatment with VX-765 does not delay the progressive deposition of Aβ in J20 mice.

**VX-765 pre-symptomatic treatment does not alter IDE and Nep.** Aβ-degrading insulin degrading enzyme (IDE) and neprilysin (Nep) were measured to assess whether VX-765 affects Aβ clearance enzymes[45,46]. *IDE* mRNA (Supplementary Fig. 9a) or protein levels (Supplementary Fig. 9b and Fig. 6e) did not change between WT and vehicle- or VX-765-treated J20 mice. Similarly, *Nep* mRNA (Supplementary Fig. 9c) and protein levels (Supplementary Fig. 9d and Fig. 6f) also remained unchanged. Since these two important Aβ-clearing enzymes are unchanged and Aβ levels are also unaffected after VX-765 treatment, it is likely that VX-765 does not act on Aβ clearance.

**Glial activation, not Aβ, associates with cognitive deficits.** Our data suggests that APP overexpression in the J20 mouse causes cognitive deficits through inflammation, rather than Aβ deposition. Indeed, there was a strong negative correlation between Iba1-positive microglial density and NOR discrimination index in the hippocampus and cortex of WT and J20 mice (Fig. 7a). There was, however, no relationship between total IL-1β levels and NOR behaviour (Fig. 7b). Neither Aβ staining density nor ELISA-measured total RIPA-soluble Aβ levels correlated with NOR discrimination index in J20 mice (Fig. 7c, d). These results suggest that microglial activation, rather than Aβ, is associated with cognitive deficits in the J20 mouse.

**Discussion**
Our data indicates that VX-765 is an effective treatment, without a sustained administration, in delaying cognitive deterioration in an AD mouse model. A 1-month pre-symptomatic administration of VX-765 can delay cognitive impairment by at least 5 months in J20 mice. Conversion of mice age and time of treatment to human age suggests that a 3–4-year pre-symptomatic treatment in humans could prevent age-dependent cognitive deficits for ~10–15 years[47]. Only a few studies implemented similar experimental paradigms[48,49], but shorter WO periods were assessed. Van Dam and colleagues observed improved acquisition and spatial memory in the Morris water maze at 3-week WO with a 2-month galantamine or memantine pre-symptomatic treatment of APP23 mice[48]. Smith and colleagues observed continued improved performance in NOR in a 1-month pre-treatment with saracatinib, but not memantine, in APP/PS1 at 3-week WO[49]. Therefore, it is impressive to observe the VX-765's disease modifying effect on preventing episodic and spatial memory deficits for up to

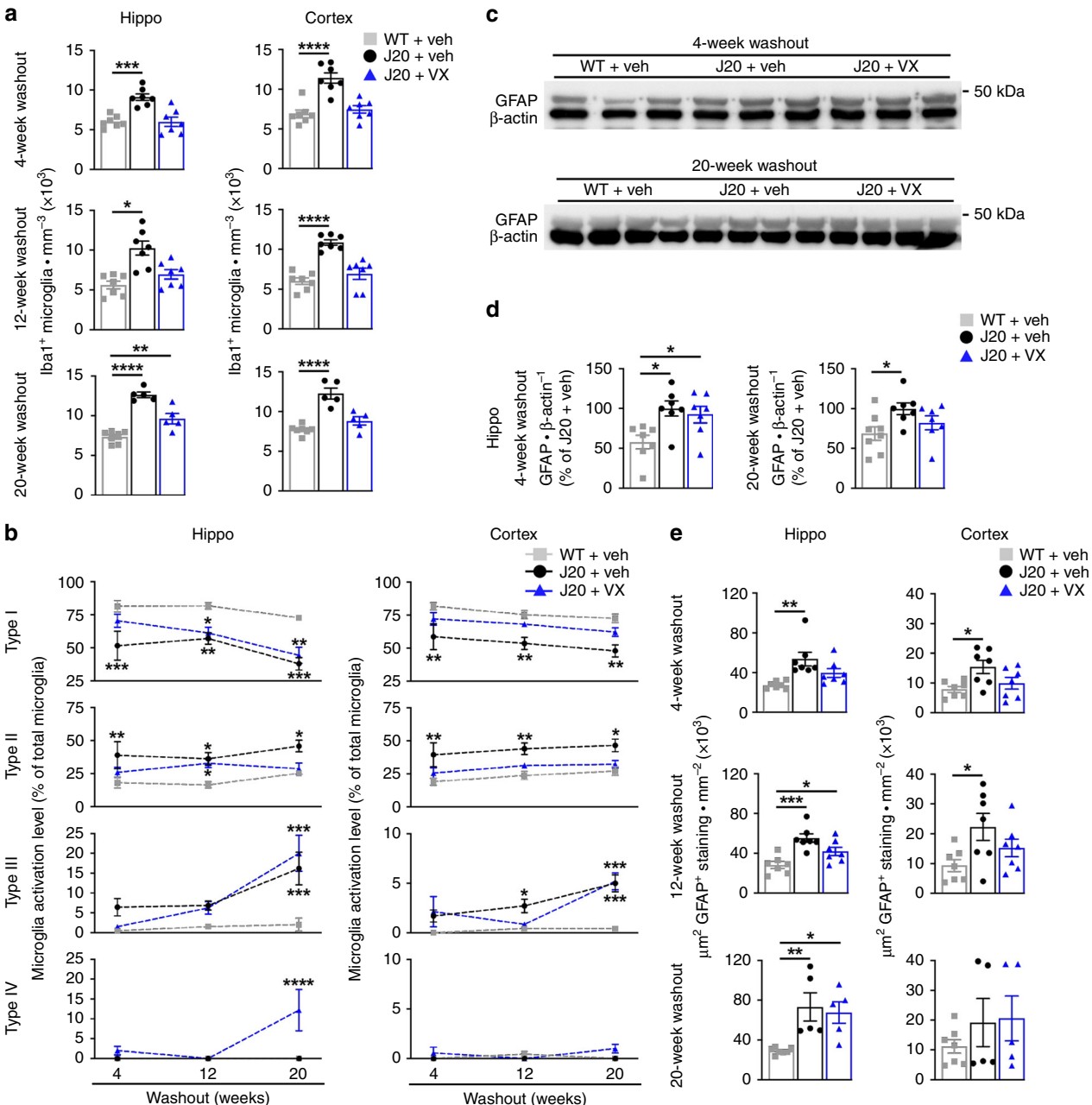

**Fig. 4 Pre-symptomatic VX-765 treatment decreases Iba1-positive microglial inflammation in J20 mice. a** Iba1-positive microglial quantification of vehicle-treated WT and J20, and VX-765-treated J20 mice from the pyramidal cell layer to the SLM in the hippocampal CA1 [hippo 4-week WO $F_{(2,18)} = 16.42$, $p = 8.74 \times 10^{-5}$; hippo 12-week WO $F_{(2,18)} = 12.25$, $p = 0.0004$; hippo 20-week WO $F_{(2,14)} = 40.35$, $p = 1.54 \times 10^{-6}$] and cortical retrosplenial and S1 area [cortex 4-week WO $F_{(2,18)} = 20.53$, $p = 2.27 \times 10^{-5}$; cortex 12-week WO $F_{(2,18)} = 25.82$, $p = 5.14 \times 10^{-6}$; cortex 20-week WO $F_{(2,14)} = 25.92$, $p = 1.96 \times 10^{-5}$, ANOVA, Dunnett's post hoc compared to WT + vehicle]. **b** Mean percentage distribution of morphological microglial subtype I [hippo treatment interaction $F_{(2,50)} = 22.47$, $p = 1.09 \times 10^{-7}$; across WO $F_{(2,50)} = 7.044$, $p = 0.0020$; cortex treatment interaction $F_{(2,50)} = 16.96$, $p = 2.38 \times 10^{-6}$], subtype II [hippo treatment $F_{(2,50)} = 12.37$, $p = 4.31 \times 10^{-5}$; cortex treatment $F_{(2,50)} = 14.66$, $p = 9.78 \times 10^{-6}$], subtype III [hippo treatment $F_{(2,50)} = 18.04$, $p = 1.26 \times 10^{-6}$; WO $F_{(2,50)} = 19.50$, $p = 5.49 \times 10^{-7}$; treatment × WO interaction $F_{(4,50)} = 5.098$, $p = 0.0016$; cortex treatment $F_{(2,50)} = 14.37$, $p = 1.18 \times 10^{-5}$; WO $F_{(2,50)} = 8.890$, $p = 0.005$, treatment × WO interaction $F_{(4,50)} = 2.822$, $p = 0.0346$, two-way ANOVA, Dunnett's post hoc compared to WT + vehicle], and subtype IV (no significant difference). In (**a**) and (**b**), $n = 7$ mice per group at 4-week WO; $n = 7$ mice per group at 12-week WO; $n = 7$ WT + veh, 5 J20 + veh, 5 J20 + VX, mice at 20-week WO. **c** GFAP and β-actin western blot in the hippocampus of three independent mice at 4- and 20-week WO. **d** GFAP western blot quantification in the hippocampus at 4-week ($n = 7$ mice per group) and 20-week ($n = 8$ WT + veh, 7 J20 + veh, 7 J20 + VX mice). WO: [4-week WO $F_{(2,18)} = 5.577$, $p = 0.0130$; 20-week WO $F_{(2,19)} = 3.629$, $p = 0.0463$, ANOVA, Dunnett's post hoc compared to WT + vehicle]. **e** GFAP positive immunostaining density in the hippocampal CA1 and retrosplenial and S1 cortex [hippo 4-week WO $F_{(2,18)} = 7.546$, $p = 0.0042$; cortex 4-week WO $F_{(2,18)} = 4.571$, $p = 0.0248$; hippo 12-week WO $F_{(2,18)} = 11.66$, $p = 0.0006$; cortex 12-week WO $F_{(2,18)} = 3.723$, $p = 0.0443$; hippo 20-week WO $F_{(2,14)} = 7.763$, $p = 0.0054$, ANOVA, Dunnett's post hoc compare to WT + vehicle]. Mouse numbers in (**e**) are the same as in (**a**). Data represents mean and s.e.m. *$p < 0.05$, **$p < 0.01$, ***$p < 0.001$, ****$p < 0.0001$.

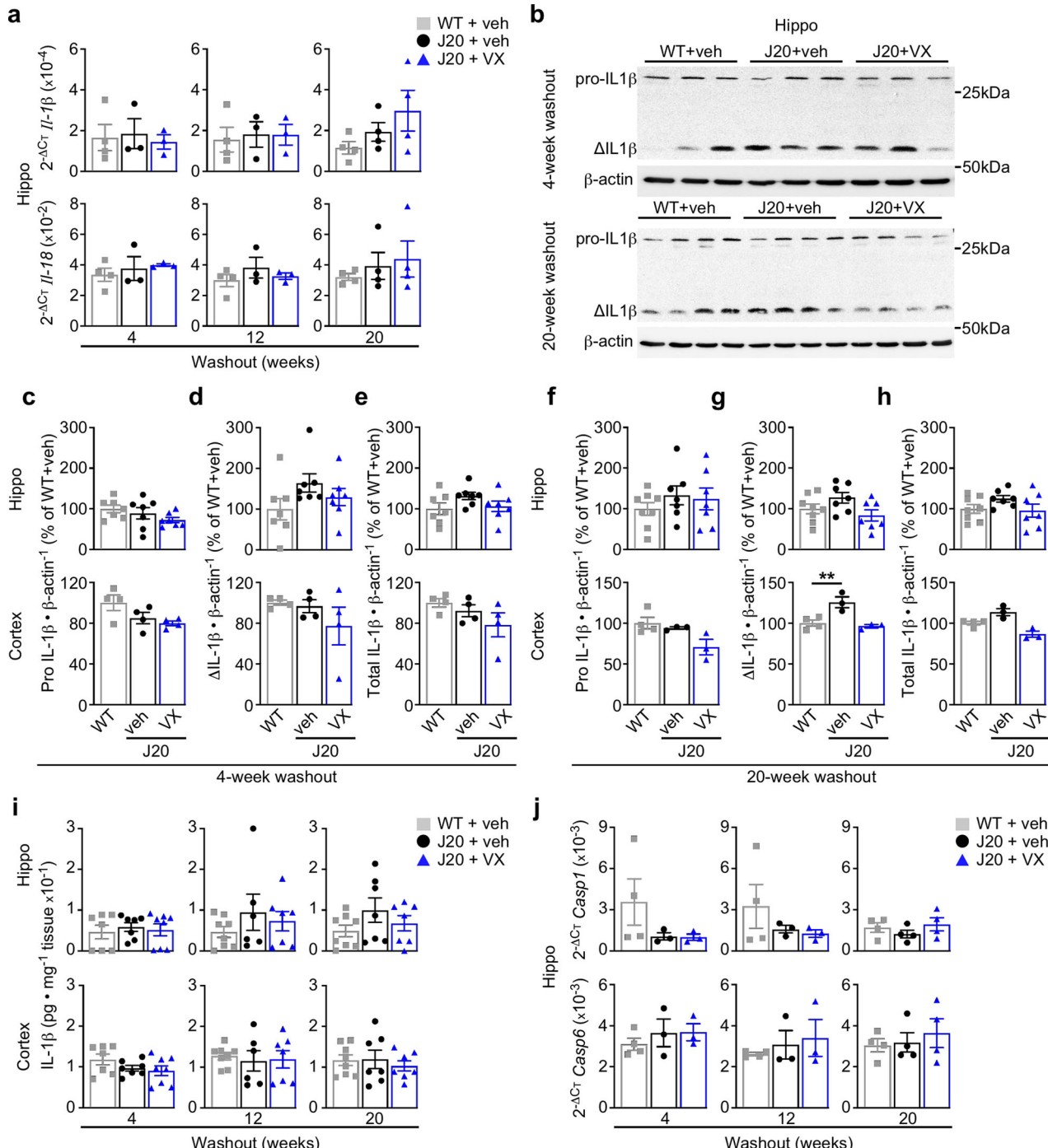

**Fig. 5 Pre-symptomatic VX-765 treatment normalizes IL-1β but does not affect inflammasome and caspase expression. a** *Il-1β* and *Il-18* mRNA levels in the hippocampus at 4-week (n = 4 WT + veh, 3 J20 + veh, 3 J20 + VX mice), 12-week (n = 4 WT + veh, 3 J20 + veh, 3 J20 + VX mice) and 20-week (n = 4 mice per group) WO. **b** IL-1β western blots showing hippocampal inactive (pro-) and active (Δ) IL-1β per group at 4- and 20-week WO. **c–h** IL-1β western blot quantification in the hippocampus and cortex comparing (**c**, **f**) pro IL-1β, (**d**, **g**) ΔIL-1β and (**e**, **h**) total IL-1β at (**c–e**) 4-week WO, and (**f–h**) 20-week WO [cortical ΔIL-1β 20-week WO F(2,7) = 12.03, p = 0.0054, ANOVA, Dunnett's post hoc compared to WT + vehicle]. In (**c–h**), n = 7 mice per group in 4-week WO hippocampus; n = 8 WT + veh, 7 J20 + veh, 7 J20 + VX mice in 20-week WO hippocampus; n = 3 per group in 4- and 20-week WO cortex. **i** ELISA measuring total hippocampal and cortical IL-1β levels at 4-week (n = 7 WT + veh, 7 J20 + veh, 8 J20 + VX mice), 12-week (n = 8 WT + veh, 6 J20 + veh, 7 J20 + VX mice) and 20-week (n = 8 WT + veh, 7 J20 + veh, 7 J20 + VX mice) WO. **j** *Casp1* and *Casp6* mRNA levels in the hippocampus at 4-week (n = 4 WT + veh, 3 J20 + veh, 3 J20 + VX mice), 12- (n = 4 WT + veh, n J20 + veh, 3 J20 + VX mice) and 20-week (n = 4 mice per group) WO. Data represents mean and s.e.m. **p < 0.01.

5 months after treatment. These results support the notion that successfully treating AD requires early intervention. Nevertheless, it is important to note that the use of VX-765 is not limited to a preventative effect. The Casp1 inhibitor, VX-765, can reverse episodic and spatial memory deficits when therapeutically given after symptom onset in J20 mice[19]. Taken together, these studies suggest that VX-765 may be beneficial in both prevention and treatment of AD.

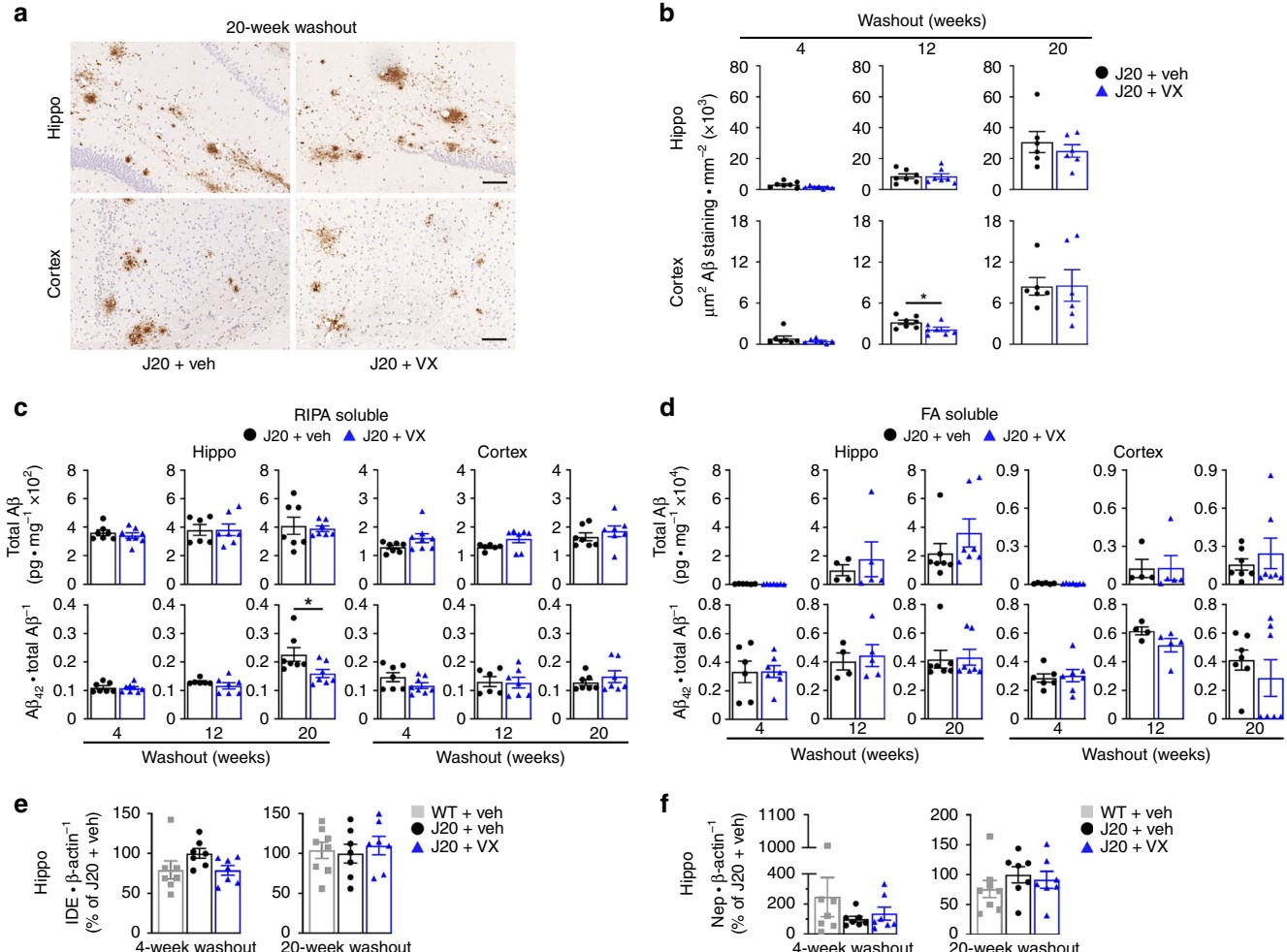

**Fig. 6 Pre-symptomatic VX-765 treatment does not affect Aβ accumulation and deposition. a** Representative Aβ micrographs of hippocampal SLM (top panels) and retrosplenial cortex (bottom panels) in vehicle- and VX-765 treated J20 mice as indicated in (**b**). Scale bar in hippo = 200 μm, cortex = 50 μm. **b** Quantitative analysis comparing Aβ immunostaining density between vehicle- and VX-765-treated J20 mice in the hippocampal pyramidal cell layer to the SLM and retrosplenial cortex at 4-week (n = 7 mice per group), 12-week (n = 7 mice per group) and 20-week (n = 6 mice per group) WO (cortex 12-week WO, p = 0.0402, two-tailed, unpaired t test). **c** RIPA-soluble and **d** FA-soluble total Aβ levels and Aβ$_{42}$/total Aβ ratio in the hippocampus and cortex (Aβ$_{42}$/total Aβ hippo 20-week WO, p = 0.0369, two-tailed, unpaired t test). In (**c**), n = 7 J20 + veh and 8 J20 + VX mice at 4-week WO; n = 6 J20 + veh and 7 J20 + VX mice at 12-week WO; n = 7 mice per group at 20-week WO. In (**d**), n = 7 J20 + veh and 8 J20 + VX mice at 4-week WO; n = 4 J20 + veh and 5 J20 + VX at 12-week WO; n = 7 mice per group at 20-week WO. **e** IDE and **f** Nep western blot quantification in the hippocampus at 4- and 20-week WO. In (**e**) and (**f**), n = 7 mice per group at 4-week WO and n = 8 WT + veh, 7 J20 + veh, 7 J20 + VX mice at 20-week WO. Data represents mean and s.e.m. *p < 0.05.

Preventing cognitive decline with VX-765 suggests that mutant APP expression in J20 mice causes cognitive impairment via Casp1-mediated inflammation. Our current and previous studies suggest the following order of pathological events: neuronal APP$^{Sw/Ind}$ expression under the PDGF-β promoter → neuronal degeneration → increased microglial activation → cognitive impairment → Aβ accumulation and deposition. Whether inflammation is controlled by VX-765 in neurons, microglia or both is debatable. Microglia are implicated in this model since J20 mice have increased Iba1-positive microglia numbers as early as 4 months of age and microglial density is strongly correlated with declining cognitive performance. VX-765 normalizes this; therefore, we can conclude that the action of VX-765 is, in part, through microglia. However, we cannot exclude the possibility that the beneficial action of VX-765 preventative treatment targets neurons. Stress by serum deprivation or overexpression of WT or mutant APP in primary human neuron cultures induces a Nlrp1–Casp1–Casp6 neurodegenerative pathway resulting in the

aggregation of several proteins including Tau, ubiquitin and microtubule-associated protein 2[16,25]. We hypothesized that neuronal Casp1 activation leads to Casp6-mediated neurodegeneration, increases Aβ production[23,50,51], and microglial activation via Il-1β[21]. The fact that Il-1β levels are extremely low in J20 mice brains (shown here and previously[19]) and in human neurons compared to easily detectable levels in activated microglia[16], mutant APP expression in J20 mice is regulated by a predominantly neuronal promoter[44,52], and cognitive function is generally a measure of neuronal function, supports VX-765's putative role in preventing neuronal degeneration.

VX-765's beneficial effect on normal mice exposed a subtle age-related cognitive decline underscoring pathways common to aging and AD, albeit accelerated in AD. VX-765 eliminated age-dependent episodic and spatial memory decline in WT mice. Recent work has shown that the age-dependent breakdown of the blood–brain barrier, leading to the influx of toxic products into the brain, can induce neuronal dysfunction and impair

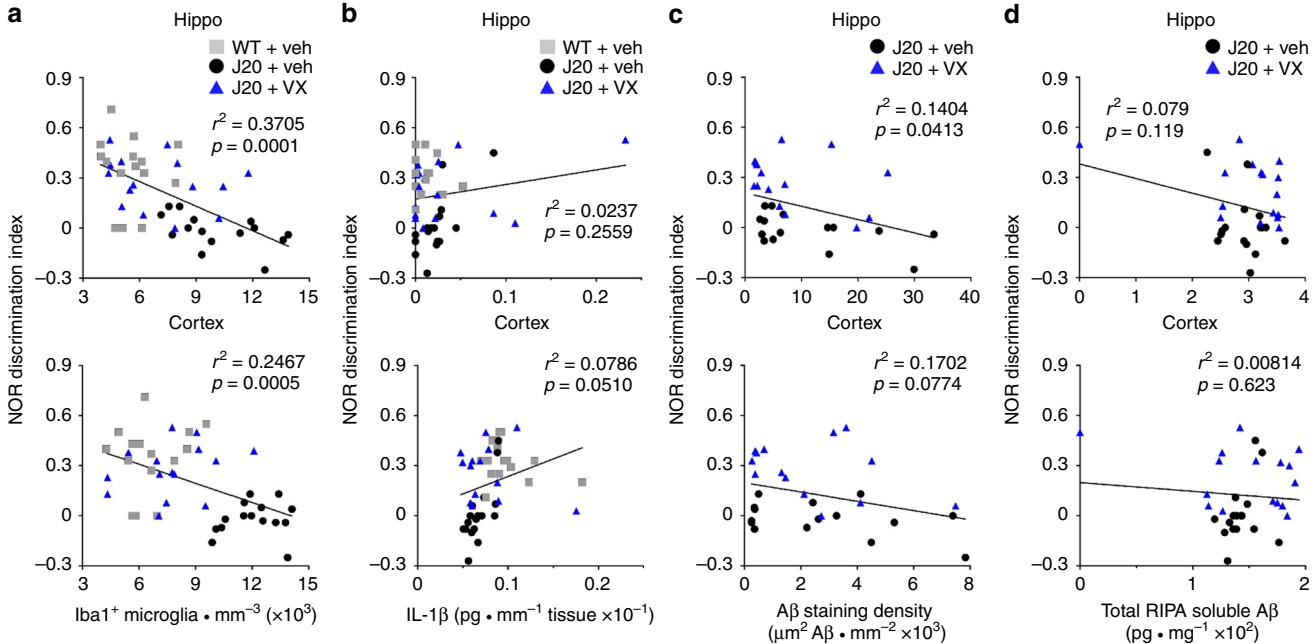

**Fig. 7 Microglial activation, not Aβ, leads to cognitive deficits in J20 mice. a–d** Scatter plot and linear regression analyses of all behaviourally-tested WT and J20 mice comparing NOR discrimination index to **a** Iba1-positive microglial numbers (hippo $r^2 = 0.3705$, cortex $r^2 = 0.2467$), **b** ELISA-measured IL-1β (hippo $r^2 = 0.0237$, cortex $r^2 = 0.0786$), **c** Aβ staining density (hippo $r^2 = 0.1404$, cortex $r^2 = 0.1702$) and **d** RIPA-soluble total Aβ (hippo $r^2 = 0.079$, cortex $r^2 = 0.00814$). In (**a**), $n = 15$ mice per group, (**b**) $n = 17$ WT + Veh, 16 J20 + Veh and 16 J20 + VX mice, (**c**) $n = 15$ J20 + Veh and 15 J20 + VX mice, (**d**) $n = 16$ mice per group.

cognition[53,54]. This can potentially lead to inflammaging, a low-grade inflammation that occurs in aging[55] and involves a microglial shift towards an immunologically primed state[56,57]. Given VX-765's effect on delaying microglial activation in J20 mice, it is not unreasonable to suggest that VX-765 acts to inhibit neurodegenerative and inflammatory pathways common in AD- and age-related cognitive decline.

The in vivo activation of inflammasomes and Casp1 is a transient process. The generation of the active, inflammasome-bound, Casp1 p33/p10 species is followed by a second self-cleavage event releasing Casp1 p20/p10 subunits from the inflammasome[58]. The p20/p10 subunits are unstable leading to their release from the inflammasome, thereby deactivating Casp1. Casp1 mRNA and protein did not change, and were unaltered by VX-765, in the J20 compared to WT brain. Furthermore, active Casp1 subunits were undetectable in brains. It is possible that (1) 8-month-old J20 brain inflammation is insufficient to detect active Casp1 subunits by western blots, (2) active Casp1 subunits are rapidly secreted and degraded in the extracellular milieu of the brain, or (3) VX-765 against Casp1 occurs in neurons where levels are very low. Only Heneka and colleagues[15] successfully showed Casp1 subunits in the APP/PS1 AD mouse model. Their mice were twice the age (16 months) of J20 mice (8 months), an age where amyloid deposits are quite extensive compared to the younger J20. Even if we could detect the active Casp1 subunits in J20 brains, we would not expect a change with VX-765, since it is a reversible inhibitor and should not stop Casp1 processing. Nevertheless, the role of Casp1 is undeniable in the J20 model since J20 on a Casp1 null background fails to develop cognitive deficits[19].

One limitation in this study is the low effect of VX-765 on IL-1β. We expected VX-765 to directly reduce IL-1β[19], yet IL-1β increased only at 20-week WO in J20 mice, and was normalized by VX-765. Possibly. IL-1β levels were low, although within the detection limit, because of the young age or lack of priming in J20 brains. IL-1β activation depends on priming that leads to the

transcriptional activation of *IL1B*[42,43]. Aβ is a priming agonist that induces inflammasome activation[15,59], but is still relatively low in young J20 mice. Alternatively, VX-765 may exert its effect on neurons, where Il-1β levels are low. Although we[19] and others[60] confirm VX-765's specificity to Casp1 against all other caspases, it is not unreasonable to suggest that Casp1 may act on many brain protein substrates other than IL-1β to mediate inflammation[61].

Aβ does not appear to be responsible for cognitive deficits in J20 mice. Despite its important delay in cognitive decline, preventative VX-765 treatment did not alter brain Aβ in J20 brains. No association was observed between soluble and insoluble Aβ levels and NOR performance in J20 mice. Therefore, our data are inconsistent with the amyloid hypothesis that presents Aβ accumulation as the primary event to neuroinflammation and neuronal dysfunction[62]. Consistent with our findings, episodic memory impairments in J20 mice develop well before Aβ accumulation[37,38,44,63]. Finally, a systematic review of transgenic AD mice corroborates no overall correlation between Aβ levels and cognitive function[64]. How these observations translate clinically remains to be seen but they are consistent with several findings. First, elderly patients with high amyloid levels do not necessarily suffer from AD[65] and, in some cases, plaque development in normal individuals is as extensive as that seen in AD patients[66,67]. Second, amyloid pathology precedes dementia by decades[68]. While some see this as evidence of causation, the possibility that amyloid deposition is merely an associated event coinciding with or exacerbating disease cannot be excluded. Finally, immunotherapies that strongly decrease Aβ levels have so far not improved AD severity, duration or progression[3,69].

In contrast, VX-765 preventative treatment delays microglial activation and suggests that altering the inflammatory milieu early is critical and can induce disease modifying effects to improve cognition. Recent clinical trials with NSAIDs have disappointedly failed to show beneficial effects[12,13], despite being

relatively successful in AD mouse models[11,13,70,71] and might raise concerns about using VX-765 against AD. NSAIDs prevented cognitive impairment in APP/PS1 AD mice, independently from Aβ levels or Iba1-positive microglial activation effects[71]. Therefore, VX-765 differs significantly from NSAIDs, notwithstanding the different targets of NSAIDs (i.e. cyclooxygenases) and VX-765 (i.e. Casp1). VX-765 can reduce microglial and astroglial activation[19], and here, we show that the effect is sustained several months after the drug removal. Fenamate NSAIDs inhibiting the Nlrp3 inflammasome and protecting against amyloid-injected rat or the AD 3x transgenic mouse[14] may also be useful in therapeutic prevention or treatment against AD.

Together, these results indicate that preventing Casp1-mediated inflammation pre-symptomatically with a VX-765 short-term treatment is sufficient to delay the onset of cognitive deficits in J20 mice. It is likely that VX-765 may prevent an early APP mutant-mediated and Casp1-dependent deleterious neuronal pathway that causes cognitive deficits and also controls subsequent microglial activation. In the future, it would be interesting to assess the molecular patterns associated with VX-765 treatment in J20 mice in hopes of identifying novel therapeutic targets against cognitive deficits.

## Methods

**Experimental design and VX-765 treatment.** All animal procedures followed the Canadian Council on Animal Care guidelines and were approved by the McGill University Animal Care committee. Male and female J20 mice (JAX Stock No. 006293, Jackson Laboratories, ME, USA) and their C57BL/6J wild-type (WT) littermates were used. Mice were group-housed between 2 and 4 mice per cage in macrolon cages under 12-h reverse light-dark cycle and controlled environmental conditions. Food and water were available ad libitum.

VX-765 (Adooq Bioscience, Irvine, CA, USA) was dissolved 5 mg • ml⁻¹ in 20% cremophor in $dH_2O$[60] (Sigma-Aldrich, ON, Canada) and administered IPWT and J20 mice received three injections per week of either 50 mg • kg⁻¹ VX-765 or 20% cremophor vehicle for 4 weeks between 2 and 3 months of age. Mice were longitudinally followed starting 4 weeks after the last injection [4-week washout (WO)], then every other month at 12- and 20-week WO (6 and 8 months of age, respectively). Intermediate 8- and 16-week WO periods were also looked at with a smaller number of mice. After each WO period, animals were randomly chosen, independent of behavioural performance, for sacrifice and assigned to either histological or biochemical analysis (Fig. 2a).

**VX-765 and VRT-043198 pharmacokinetics.** Plasma, whole brain and CSF concentrations of VX-765 and its active metabolite, VRT-043198, were determined by LC-MS/MS at Paraza Pharma (Montreal, QC, Canada). Fourteen C57BL/6J female mice received a single IP dose of 50 mg · kg⁻¹ VX-765 and 2 mice per time point were euthanized at 0.25, 0.5, 1, 3, 6, 8 and 24 h post-injection. Mice were isoflurane anesthetized to collect CSF and perform a cardiac puncture followed by whole body perfusion with PBS. Plasma samples were extracted 1:1 in spiking solution and internal standard solution containing 0.1 μM glyburide in acetonitrile. Brain and CSF samples were extracted 1:1 in spiking solution and internal standard solution containing 0.2 μM labetalol in 50:50 methanol:acetonitrile. Samples were centrifuged at $4000 \times g$ for 10 min at 4 °C, the plasma supernatant diluted 1:1 and the brain and CSF supernatant diluted 1:2 in $H_2O$, and 5 μl was injected into a Quantis LC-MS/MS system for analysis.

**Behavioural analysis.** Open-field and novel object recognition (NOR) were measured at all WO periods. The Barnes maze was assessed at 12- and 20-week WO. Open field: Mice were placed in an open-field chamber, composed of a plexiglass box with no ceiling and white floor, and allowed to explore for 5 min. NOR: Mice were pre-exposed to two identical objects in the open-field chamber for 5 min. Two hours after pre-exposure, mice were placed back into the chamber and exposed to one familiar object and one novel object for 5 min (test phase). Mice were exposed to a specific object only once across multiple test sessions. Barnes maze: Learning acquisition (days 1–4)—each mouse was trained to find the target and enter the escape box within 180 s. Four trials per day, ~15 min apart, were performed for 4 consecutive days. Probe (day 5)—a single, 90 s trial was performed on each mouse to measure primary latency and number of errors to reach the blocked target. Target hole location was changed between the two separate test sessions. Behavioural scoring was blinded to mouse genotype and treatment.

**Immunohistochemistry and quantification.** Mice for immunohistochemistry were isoflurane anesthetized and perfused with ice-cold 4% paraformaldehyde in 0.1 M

PBS (Sigma-Aldrich). Brains were post-fixed overnight in 10% neutral-buffered formalin (ThermoFisher Scientific, ON, Canada) and transferred to 70% ethanol for paraffin embedding and sectioning at 4 μm. Tissue slides underwent heat-induced antigen retrieval with citrate or EDTA buffer. Immunostaining was performed using the Dako Autostainer Plus slide processor and EnVision Flex system (Dako, ON, Canada). Sections were peroxidase treated, blocked in Serum-Free Protein Block, and immunostained with the following antibodies: 1:2000 rabbit anti-Iba1 (Wako, VA, USA), 1:500 rat anti-CD68 (clone FA-11, Bio-Rad, Hercules, CA, USA), 1:8000 rabbit anti-GFAP (Dako) and 1:2000 rabbit anti-Aβ₁₋₄₀ (F25276, laboratory developed). CD68 antibody required an additional rabbit anti-rat IgG/HRP secondary reagent (Dako) incubation. Immunoreactivity was revealed with HRP-conjugated secondary antibody and diaminobenzidine and hematoxylin counterstain. Sections were digitally scanned with MIRAX SCAN (Zeiss, ON, Canada) and used for subsequent analyses. Representative images have only been cropped and did not undergo any post-processing.

Iba1-positive microglia were quantified using a modified version of an area-specific counting frame[72]. The number of Iba1-positive cells per $mm^3$ in the anterior CA1 region and cortex was estimated across four sections, spaced 60 μm apart. Microglial activation level was determined based on morphological measurements[19,41]. Aβ₁₋₄₀ and GFAP immunostaining density was measured across four sections using ImageJ software (NIH, Bethesda, MD, USA). Quantification was blinded to treatment and genotype.

**Protein extraction.** Mice for biochemical analyses were isoflurane anesthetized, sacrificed by cervical dislocation, and the hippocampus and cortex dissected out and immediately frozen on dry ice. Brain tissue was homogenized in radio-immunoprecipitation assay (RIPA) buffer (50 mM Tris-HCl pH: 7.4, 1% NP-40, 0.25% Na-deoxycholate, 150 mM NaCl, 1 mM EDTA, with protease inhibitors), and supernatants placed in Laemmli (5% 2-β mercaptoethanol with protease inhibitors).

For Casp1 positive control, J774A.1 murine macrophages (ATCC, Manassas, VA, USA) were grown in Dulbecco's modified Eagle medium (DMEM) (ThermoFisher Scientific) supplemented with 10% foetal bovine serum (Wisent, QC, Canada). Twenty-four hours before treatment, cells were plated in 6-well plates, washed with PBS, and then treated with 200 ng/mL lipopolysaccharide (LPS) (Millipore, ON, Canada) in DMEM for 4 h. Nigericin (Sigma) or an equivalent volume of DMSO/ethanol (1:1) was then added to the media to reach a final concentration of 13.8 μM and incubated an additional 2 h. Culture medium was collected, centrifuged at $94 \times g$ for 5 min to remove cellular debris, and proteins from 500 μl of supernatant were precipitated with trichloroacetic acid and resuspended in 40 μl of loading buffer. Cells were washed with PBS and harvested in RIPA buffer.

**Western blot and quantification.** Twenty to sixty micrograms of protein or 16 μl of precipitated protein from culture medium were separated on SDS-PAGE or 4–12% NuPAGE Bis-Tris gels (Invitrogen, Carlsbad, CA, USA) with MES buffer (Invitrogen) and transferred onto PVDF membranes (Bio-Rad) according to the manufacturer's instructions. Membranes were probed with the following antibodies: 1:1000 mouse anti-Aβ₁₋₁₆ (6E10) (BioLegend, CA, USA), 1:2000 rabbit anti-APP C-terminus (A8717) (Sigma-Aldrich), 1:1000 rabbit anti-neprilysin (Nep, Abcam, ON, Canada), 1:500 rabbit anti-insulin degrading enzyme (IDE, Abcam), 1:3000 rabbit anti-GFAP (Dako), 1:1000 rabbit anti-IL-1β (Abcam), 1:1000 rat anti-Caspase-1 (clone 4B4.2.1, kindly provided by Genentech, San Francisco, CA, USA) and 1:1000 mouse anti-β-actin (Sigma-Aldrich). Immunoreactivity was revealed with 1:5000 HRP-conjugated secondary antibodies (Jackson ImmunoResearch, West Grove, PA, USA or GE Healthcare BioSciences, NJ, USA) and detected with ECL (GE Amersham, ON, Canada). Signals were visualized with the ImageQuant LAS 4000 imaging system and densitometric analyses were performed with Image Gauge analysis software 3.0 (Fujifilm USA, NY, USA). Protein extracts from three independent mice in each group were run together on each blot. Protein levels were normalized to β-actin and expressed as a percentage relative to the vehicle-treated WT group, except for APP which was compared to the vehicle-treated J20 group. Representative full blots are shown in the Source data file.

**ELISA.** ELISA samples were prepared in RIPA described above. RIPA-insoluble pellets were further extracted in 70% formic acid (FA) in $dH_2O$. FA was evaporated in a speed vacuum, and the resulting pellet was solubilized in 200 mM Tris-HCl, pH 7.5. IL-1β and Aβ levels (which include Aβ₃₈, Aβ₄₀ and Aβ₄₂) were measured using Meso Scale Discovery ELISA kits (Rockville, MD, USA). Standards and samples were prepared according to the manufacturer's protocols and run in duplicate.

**RNA extraction and real-time PCR.** RNA was extracted with Qiazol and purified using the miRNeasy mini kit (Qiagen, CA, USA). RNA quality and quantity were determined using a spectrophotometer (DS-11 FX+, DeNovix, DE, USA) and RNA was reverse transcribed using First Strand cDNA Synthesis Kit (Qiagen) following the manufacturer's protocol.

Real-time PCR experiments were performed with SYBR Green Taq Mastermix (Quanta BioSciences, MD, USA) in a real-time cycler ABI 7500 Fast Block (Applied

Biosystems, CA, USA). Primer sequences are described in the supplementary material. Casp1 (MP201790) and Casp6 (MP201796) primers were purchased from Origene (MD, USA). Results are presented as individual data points using the $2^{-\Delta Ct}$ method[73].

**Statistics and reproducibility**. Mouse behavioural experiments were performed on five independent cohorts generated at different times. Each cohort produced similar results with low variability between experiments. Post-mortem histological and biochemical analyses were performed on all cohorts. Immunohistochemistry was performed within each of the five independent experiments and produced similar results when repeated across each experiment. Western blots, ELISA and PCR were performed all together after all behavioural testing was finished and when tissues from all WO periods were available.

All mice are indicated as individual data points in the figures and values are expressed as mean and s.e.m. Statistical analyses for behaviour were performed with regular or repeated-measures two-way ANOVA and either Dunnett's post hoc compared to WT + vehicle (used between treatment groups within WO periods) or Bonferroni's post hoc (used between same treatment group across different WO periods). Histological and biochemical analyses were performed with unpaired Student's $t$ test or ANOVA with Dunnett's post hoc compared to WT + vehicle or J20 + vehicle. Statistical significance for group comparisons was set to $p < 0.05$. Pearson correlation and linear regression was performed between behavioural performance and Iba1-positive microglial density, IL-1β levels, Aβ staining density and total RIPA-soluble Aβ levels. Given the numerous correlations, a Bonferroni's adjustment was used to reduce the significance level to $p < 0.01$[74]. All statistical analyses were conducted using GraphPad Prism 8 software (GraphPad Software, CA, USA). Total numbers ($n$), F and p values are described in the figure legends.

**Reporting summary**. Further information on experimental design is available in the Nature Research Reporting Summary linked to this paper.

## Data availability

The authors declare that all data within the manuscript and its supplementary file in this study are available. Microscope slides and/or digital scans of immunohistological staining, including access to software to view scans, are available upon request and provision of a depository site with sufficient memory to accept the files. Any additional information is available upon request. Source data are provided with this paper.

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

## Acknowledgements

We would like to thank Marie-Lyne Fillion for helping with the management of the project and keeping track of the mice over the course of this study. This work was supported by funds from the Canadian Institutes of Health Research (CIHR) 201610PJT-377052-PJT-CFAF-45097 Ref # 153097, and the JGH Foundation to A.L.B.

## Author contributions

J.F.: animal behaviour and analyses, immunohistochemistry analyses, ELISA, designed the experimental paradigm, wrote and revised the paper and made figures. A.N.: western blot analyses, prepared figure panels, wrote methods and revised the paper. B.F.: PCR, wrote methods and revised the paper. O.B.: revised the paper. A.L.B.: conception of the experimental idea and design, supervision of data collection and analysis and wrote and revised the paper.

## Competing interests

The authors declare no competing interests.
