## [Peer Review File · Nature Communications]

Reviewers' comments:

Reviewer #1 (Remarks to the Author):

General

In this manuscript, Flores et al describe the effect of VX765, a caspase-1 inhibitor, on learning and memory, as well as neuropathology in a model of cerebral amyloidosis (J20 mice). The key findings suggest that VX765 treatment delays the onset of learning and memory deficits and suppresses neuroinflammatory processes without altering beta-amyloid deposition in this model. One beauty of the study is the introduction of washout periods in order to observe the lasting effects of treatment. In general this is a well conducted work, which is certainly timely and of interests to the broader readership of Nat Comm.. I had several comments and concerns, most of them of minor nature, which should be considered.

Detail

Line 183

In the view of this reviewer, the phagocytic nature of a microglial cell cannot be determined by morphological categorization (in the case of the present paper class IV microglia). In order to substantiate the claim that VX765 promotes phagocytosis, the authors should provide in vivo microglia uptake experiments.

Line 188-189

This statement does not reflect the data given in Figure 3 d as neither at 4 nor at 20 weeks WO period, VX exerts a significant reduction of GFAP levels. The authors may consider to increase the number of observations in order to re-test for significance (since a tendency is visible).

Line 192-195

The authors also need to consider that the observed differences may results from the comparison of different time periods: 12 weeks vs 20 weeks WO.

Line 202-204

Fig. 4 a does not depict data from 16 week WO. Please comment.

Line 210-211

While the data shown in 4g demonstrate a VX765-associated reduction of active interleukin-1 beta levels, it seems unclear whether this in both, the hippocampus and neocortex, reached the level of statistical significance. Please comment.

Line 229-230

mRNA levels of Caspase1 and 11 should be complemented by Western blot analysis of the respective pro- and cleaved forms.

Line 260

The authors state: "VX-765 did not affect 261 Ab degrading enzyme levels in J20 mice, suggesting that VX-765 does not act on Ab clearance".

The authors need to reconsider this statement as there are several mechanisms of amyloid beta clearance that they have not tested for including but not restricted to (I) further degrading enzyme activity e.g. ACE, ECE etc. (II) microglial phagocytosis (III) glymphatic drainage.

Reviewer #2 (Remarks to the Author):

This report aimed to test the effects of pre-symptomatic administration of a caspase-1 inhibitor to a mouse model of Alzheimer's disease. This is of interest to the research community working on neurodegenerative disease and targeting inflammation. The experiments are carefully conducted, well presented, and the paper is well written. I have some points the authors may consider:

1. On Page 8, lines 143/144, the sentence 'Together these preclinical results indicate that a one month pre-symptomatic treatment with VX-765 could prevent cognitive deficits in humans' is too bold based on the mouse data presented here and should be modified appropriately.
2. The data on IL-1b are somewhat unclear. If the caspase-1 inhibitor is effective then Pro-IL-1b processing will be inhibited. However, mature IL-1b is still present. If IL-1b levels are too low to allow an accurate measure then the authors could look at IL-18 processing, or better, caspase-1 processing, which can be measured by western blot.

Reviewer #3 (Remarks to the Author):

In this manuscript, the authors examine the effects of transient administration of the caspase-1 inhibitor VX-765 for one month in pre-symptomatic (2-month) J20 mice on the subsequent development of cognitive deficits, microglial activation, cytokines and amyloid deposits. They found that cognition was delayed by ~5 months, microglial activation by ~3 months, and IL1-beta activation by ~5 months, but no effect on amyloid deposits. They conclude that inhibiting caspase-1 blocks an effect of mutant APP on neurological function, possibly by dampening an inflammatory pathway.

This is a carefully done study, but its impact is limited in the context of what we know from human clinical trials.

1. The central concern pertains to the fact that the effects of NSAIDs in APP transgenic mice were first reported 20 years ago, and are very similar to those of VX-765, which include lowering microglial and IL1-beta activation (Lim et al 2000), and improving cognition and synaptic plasticity (Kotilinek et al 2008). Sadly, NSAIDs have not been effective in preventing or treating AD (Meyer et al 2019). Meyer et al concluded, "This work has left us with extreme pessimism regarding any possible role of NSAIDs in AD prevention." Therefore, in order to advance our understanding of the role of caspase-1 as a target for AD prevention, it is important to know how and why VX-765 differs from NSAIDs in mitigating APP-mediated abnormalities.

This concern notwithstanding, the data are carefully collected, analyzed, and presented, on the whole. Addressing the following would enhance the clarity and accuracy of the paper.

2. What is the expected brain concentration of VX-765 at the dosing regimen used here? Previous publication from this group measured the concentrations in mouse cortex and hippocampus, at 50 mg/kg. This could be added as a comment in the description of the study. "Mice were dosed at 50 mg/kg....expected brain concentrations of xx uM(ref)..."

3. Since washout and continuation of effect after stopping treatment is an important point of this paper, the half-life of VX-765 (and perhaps the active metabolite VRT-043198?) in mouse brains should be reported. If it has been reported elsewhere, please provide reference similar to the expected brain concentration of compound as above.

4. Page 5, line 93: Please define NOR at first instance of use.

5. Page 6, lines 103-110: This discussion seems to describe improved performance in NOR testing for VX-765 treated WT mice after 12-week WO as compared to the initial test at 4 months of age. At the same time, the NOR performance decreased in vehicle treated mice between 12-20 months

of WO, but did not decrease in VX-765 mice. In Fig 1b, it almost appears as if there is improvement in the WT+VX treated group from 4 to 12 weeks that is then sustained at 20 weeks, even though the performance of the WT+veh group declines over the same period. This should be commented on.

6. Page 7, lines 127-129 and Fig 1e: There was a significant difference in primary latency between the WT+veh and WT+VX groups. Can this be explained?

7. Page 15, lines 308-313 & Page 16, line 328: In the discussion section, the authors talk about the potential for VX-765 to treat neurodegeneration, yet provide no evidence for changes in neurodegeneration in the mice. The discussion should be restricted to only claim the compound can reduce inflammatory pathways and improve cognition of these mice. It is possible to hypothesize that this could affect neurodegeneration, but no evidence has been provided that this is true.

Response to Reviewers' comments:

The authors thank the reviewers for the careful revision of our manuscript. We have addressed all the comment as described below and believe that the revisions have increased the value of the paper. Because of COVID-19, it became increasingly difficult to perform experiments in the laboratory since the Institute closed its doors for several months in order to protect its employees. Despite these restriction, we believe we have been able to address all the reviewer's concerns. Our response to each concern or critique and our revision is outlined below.

Reviewer #1 (Remarks to the Author):

General

In this manuscript, Flores et al describe the effect of VX765, a caspase-1 inhibitor, on learning and memory, as well as neuropathology in a model of cerebral amyloidosis (J20 mice). The key findings suggest that VX765 treatment delays the onset of learning and memory deficits and suppresses neuroinflammatory processes without altering beta-amyloid deposition in this model. One beauty of the study is the introduction of washout periods in order to observe the lasting effects of treatment. In general, this is a well conducted work, which is certainly timely and of interests to the broader readership of Nat Comm. I had several comments and concerns, most of them of minor nature, which should be considered.

Detail

Line 183: In the view of this reviewer, the phagocytic nature of a microglial cell cannot be determined by morphological categorization (in the case of the present paper class IV microglia). In order to substantiate the claim that VX765 promotes phagocytosis, the authors should provide in vivo microglia uptake experiments.

We agree that morphological characteristics cannot definitely determine the state of the microglia. We are not familiar with a technique test the phagocytic activity of microglia in vivo that would not risk activating the microglia indirectly. Therefore, we performed immunohistochemistry with CD68, a marker of phagocytic microglia and macrophages. CD68 staining in the same tissue that was used to assess the morphology of microglia was sparse and did not correlate with type IV microglia in VX-765-treated J20 mice. Nevertheless, we only saw CD68 in the VX-765-treated mice: four out of the five VX-765 treated J20 mice showed positive CD68 staining, but no staining was found in any vehicle-treated J20 mouse. Unfortunately, these rare CD68 positive staining could not be quantified with certainty and the effect was too small to infer that phagocytosis is involved.

Therefore, we have removed microglial phagocytosis as a primary mechanism of action of VX-765. We re-wrote this section by removing the statement that VX-765 "promotes microglial phagocytosis". Instead, we made more general conclusions about VX-765's ability to delay inflammation. The new results are presented on pages 10-11 of the manuscript and in Supplemental Figure 4.

Line 188-189: This statement does not reflect the data given in Figure 3d as neither at 4 nor at 20 weeks WO period, VX exerts a significant reduction of GFAP levels. The authors

may consider increasing the number of observations in order to re-test for significance (since a tendency is visible).

In line 188-189 we had written “Hippocampal GFAP protein levels (Fig. 3c now 4c) increased in vehicle-treated J20 compared to WT mice at 4- and 20-week WO but were normalized with VX-765 treatment only at 20-week WO (Fig. 3d now 4d).” In figure 3d, we observe a statistically significant increase of hippocampal GFAP in J20 and in J20 + VX compared to WT at the 4 week WO. However, at 20 wk WO, while there is a significant increase of GFAP between WT and J20, there is no significant difference between WT and J20 + VX”. Therefore, we believe that the sentence correctly described the data.

Please note that we also conducted immunohistochemical analysis of GFAP (Fig. 4e and Supplementary Figure 5). The results showed increased GFAP immunopositivity in the vehicle-treated J20 hippocampus and cortex at 4-wk WO, and in the hippocampus at 8-wk WO that was normalized in VX-765-treated J20. The VX-765 normalization of GFAP levels was lost at 12-, 16- and 20-wk WO, which is not unexpected given the removal of the VX-765 for 3-5 months, respectively.

Line 192-195: The authors also need to consider that the observed differences may results from the comparison of different time periods: 12 weeks vs 20 weeks WO.

We wrote” The discrepancy between the western blot and immunostaining data is possibly due to differences in the regions of interest analyzed, or the dilution of GFAP with other cell proteins during extraction.”

This sentence referred to the comparison of WB (Fig. 4d) and IHC (Fig. 4e) at 4- and 20-week WO. At 4-week WO, the level of GFAP in J20 + VX is significantly different from the WT by WB, whereas it is not by IHC. At 20 week WO, the GFAP levels are significantly different between WT and J20 in both WB and IHC. However, the levels of GFAP are not significantly different between WT and J20 + VX in WB but are in IHC. Our comment referred to these difference. We have revised line 192 to 195 to clarify the comparison.

Line 202-204: Fig. 4a (now 5a) does not depict data from 16 week WO. Please comment.

The IL-1 β and IL-18 mRNA level data for the 8 and 16 week WO were presented in Supplementary Figure 6a. The reason for separation of the data at 4-, 12-, and 20-week and at 8- and 16-week WO was explained in the first paragraph of the result section. “ Mice were behaviourally assessed at 4 months of age, four weeks after their last injection, when they normally exhibit cognitive deficits and hyperactivity, and again at 6 and 8 months of age (4-, 12-, and 20-week washout [WO]) (Fig. 1a). To ensure a complete analysis of the J20’s behavioural progression, a smaller number of mice were also behaviourally assessed at 8- and 16-week WO. At each WO period, a subset of animals was euthanized for post-mortem analyses.”

Line 210-211: While the data shown in 4g (now 5g) demonstrate a VX765-associated reduction of active interleukin-1 beta levels, it seems unclear whether this in both, the hippocampus and neocortex, reached the level of statistical significance. Please comment.

We observed a statistically significant difference between WT and vehicle-treated J20 only in the cortex (Fig. 5g, bottom panel), but not in the hippocampus (Fig. 5g, top panel) at the 20-week washout but not the 4-week washout (Fig. 5d). No differences were observed in Pro-IL1 β

or total IL-1 β cortex or hippocampus (Fig. 5c, e, f, h). The increase of cleaved (Δ)-IL1 β in the vehicle-treated J20 cortex compared to WT disappeared in the VX-765-treated J20. We had discussed the issue with IL-1 β levels in the Discussion section – see last paragraph on page 19 as follows. “One limitation that is important to consider in this study is the low effect of VX-765 on IL-1 β . It would be expected that VX-765-induced Casp1 inhibition would directly reduce IL-1 β ¹⁹, yet we found only substantial evidence to support that VX-765 normalized IL-1 β in J20 mice at 20-week WO. There are a few possible explanations for this. First, measured IL-1 β levels were low although within the detection limit and gradually increased with age. Notably, IL-1 β activation depends on priming that leads to the transcriptional activation of *IL1B*^{39,40}. Without priming, IL-1 β levels remain constitutively low. It is possible that J20 mice, which did not receive a priming step, did not reach the transcriptional threshold to produce adequate IL-1 β to detect differences between groups. A β is a potential priming agonist that induces inflammasome activation^{15,58}, but the J20 mice are relatively young (4-8 months of age) and only at the beginning stages of A β accumulation. Alternatively, VX-765 may exert its effect on neurons, as discussed above. Although we¹⁹ and others⁵⁹ confirm VX-765’s specificity to Casp1 against all other caspases, it is not unreasonable to suggest that Casp1 may act on many brain protein substrates other than IL-1 β to mediate inflammation⁶⁰.”

Line 229-230: mRNA levels of Caspase1 and 11 should be complemented by Western blot analysis of the respective pro- and cleaved forms.

We did not perform Caspase-11 WB because in our previous paper (Flores et al., Nature Communications 2018), we showed that the for VX-765 IC50 for Casp11 was over 100 fold compared to Casp1. Combining this with the lack of increased Caspase-11 mRNA levels in J20 mice at a time where behavioral deficits were prevented with VX-765 early treatment, it is unlikely that the beneficial effect of VX-765 on behavior occurs through Caspase-11.

We performed a western blot with anti-Casp1 antibodies capable of detecting full-length Casp1 and p33 and p20 active subunits. We chose the latest time point so that we would have a chance of seeing the active subunits if present because caspase active subunits are notoriously difficult to detect in brain tissues. We detected the full-length Casp1 in both WT and J20 cortex and amounts were equivalent in vehicle and VX-765-treated animals. We did not detect the p33 and p20 subunits of processed Casp1. This does not entirely surprise us, as shown in our positive control using a microglial cell line, these subunits are secreted rapidly. In the brain, they may be degraded rapidly upon secretion. We know of only one group, that of Heneka, who was able to detect active Casp1 subunits in brain of APP/PS1 mice. The mice were much older (16 months of age) than ours (8 months of age). Therefore, their ability to detect active Casp1 subunits could be because they had reached extensive microglial activation and Casp1 processing in these animals. We have added our results to the paper in Supplemental Fig. 6f and 6g and explained the results on page 13 in the result and discussion section.

Line 260: The authors state: “VX-765 did not affect A β degrading enzyme levels in J20 mice, suggesting that VX-765 does not act on A β clearance”. The authors need to reconsider this statement as there are several mechanisms of amyloid beta clearance that they have not tested for including but not restricted to (I) further degrading enzyme activity e.g. ACE, ECE etc. (II) microglial phagocytosis (III) glymphatic drainage.

We agree with the reviewer. We have revised this section to indicate that we only looked at IDE and Nep. However, since we do not see any change in A β levels, it is likely that there is no change in A β clearance. We have changed the concluding sentence of this section to “ Since

these two important A β -clearing enzymes are unchanged and A β levels are also unaffected with VX-765 treatment, it is likely that VX-765 does not act on A β clearance”

Reviewer #2 (Remarks to the Author):

This report aimed to test the effects of pre-symptomatic administration of a caspase-1 inhibitor to a mouse model of Alzheimer's disease. This is of interest to the research community working on neurodegenerative disease and targeting inflammation. The experiments are carefully conducted, well presented, and the paper is well written. I have some points the authors may consider:

1. On Page 8, lines 143/144, the sentence 'Together these preclinical results indicate that a one month pre-symptomatic treatment with VX-765 could prevent cognitive deficits in humans' is too bold based on the mouse data presented here and should be modified appropriately.

We agree with the reviewer and have revised this conclusion to indicate the uncertainty in going from mouse to human. The sentence has been changed to “Together, these preclinical results raise hope that a pre-symptomatic treatment with VX-765 might prevent cognitive deficits in humans.”

2. The data on IL-1 β are somewhat unclear. If the caspase-1 inhibitor is effective, then Pro-IL-1 β processing will be inhibited. However, mature IL-1 β is still present. If IL-1 β levels are too low to allow an accurate measure then the authors could look at IL-18 processing, or better, caspase-1 processing, which can be measured by western blot.

Even though we mentioned in this paragraph that IL-1 β levels were on the low end of the detection limit, they were still within the standard curve determined with the positive control in the ELISA assay, so the data is accurate. It is just that the levels of IL-1 β are low. We changed the sentence to indicate that IL-1 β levels were low although accurately detectable.

We agree that the IL-1 β results were unexpected at first. However, once we started thinking about them, they do make sense. First, that high IL-1 β levels were not observed in the mice cortex and hippocampus is probably due to the fact that the mice are still considerably young. What we observed is that the IL-1 β mRNA levels did not increase significantly in the hippocampus and cortex of J20 mice at either 4, 8, 12, or 20 week WO, which represents mice of 4, 5, 6, and 8 months of age, respectively (Figure 4a and Supplementary Figure 5a). At 16 week WO (7 months of age), we do observe a significant increase in vehicle-treated J20 hippocampal IL-1 β mRNA levels (Supplementary Fig. 5a), which is normalized in VX-765-treated J20 mice. Therefore, at this specific time point, we have evidence that Casp1 has been inhibited by VX-765. That evidence is not observed in the other WO time points simply because there was no increased IL-1 β in J20. The discussion's sixth paragraph highlights the limitations of our studies on IL-1 β .

Mature IL-1 β is always expressed at a certain basal level in the brain and this is easily detected with western blots and with ELISA. One of the expected triggers of increased IL-1 β and IL-1 β processing in AD mouse model is through A β -mediated Nlrp3/Casp1/IL1 β in microglia. However, in the J20 mouse model, A β does not accumulate significantly in brain before 5 months of age

and is still relatively low until 8 months of age (Figure 5 b, c, and d). We had proposed this explanation in the sixth paragraph of the discussion.

We disagree that detection of Casp1 processing would confirm specific drug targeting. The VX-765 is a reversible inhibitor of active Casp1 and will prevent its activity but not its processing. However, we agree that showing the presence of processed Casp1 in J20 would further support this pathway. We and others have historically been unable to conclusively and reproducibly detect active Casp1 in mouse brain by western blotting. We find that many other papers using the VX-765 also did not show Casp1 processing, indicating this to be a general problem in research. Furthermore, companies that sell the Casp1 antibodies told us that even in pure macrophages, Casp1 is hard to detect. Consequently, it will be even more difficult, if not impossible, to detect processed active Casp1 in the brain. We tried many available antibodies and using proper negative controls (knock out Casp1 mouse brain) and positive controls (LPS-stimulated microglial cell lines and commercially available recombinant Casp1). We failed to convincingly detect active Casp1 subunits in mouse brain although we can easily detect secreted Casp1 subunits from the LPS/nigericin-treated cell line. We also tried immunohistochemistry and also found false positives in the mouse Casp1 KO brains. However, we are aware that the laboratory of Dr Heneka shows full length and processed Casp1 in the APP/PS1 old mice as indicated above. Possibly, our mice are still too young to show detectable active Casp1 by western blotting.

We have however published other evidence confirming the involvement of Casp1 in the J20 mouse model. The most convincing is in Flores et al., Nat Comm 2018, where we showed that the Casp1 null J20 are protected from memory deficits, inflammation and A β accumulation. Consistent with these findings, Casp1 siRNA protected primary human neurons from neurodegeneration.

Reviewer #3 (Remarks to the Author):

In this manuscript, the authors examine the effects of transient administration of the caspase-1 inhibitor VX-765 for one month in pre-symptomatic (2-month) J20 mice on the subsequent development of cognitive deficits, microglial activation, cytokines and amyloid deposits. They found that cognition was delayed by ~5 months, microglial activation by ~3 months, and IL1-beta activation by ~5 months, but no effect on amyloid deposits. They conclude that inhibiting caspase-1 blocks an effect of mutant APP on neurological function, possibly by dampening an inflammatory pathway. This is a carefully done study, but its impact is limited in the context of what we know from human clinical trials.

1. The central concern pertains to the fact that the effects of NSAIDs in APP transgenic mice were first reported 20 years ago, and are very similar to those of VX-765, which include lowering microglial and IL1-beta activation (Lim et al 2000), and improving cognition and synaptic plasticity (Kotilinek et al 2008). Sadly, NSAIDs have not been effective in preventing or treating AD (Meyer et al 2019). Meyer et al concluded, "This work has left us with extreme pessimism regarding any possible role of NSAIDs in AD prevention." Therefore, in order to advance our understanding of the role of caspase-1 as a target for AD prevention, it is important to know how and why VX-765 differs from NSAIDs in mitigating APP-mediated abnormalities. This concern notwithstanding, the data are carefully collected, analyzed, and presented, on the whole. Addressing the following would enhance the clarity and accuracy of the paper.

The reviewer correctly points out the disappointing effect of non-steroidal anti-inflammatory drugs (NSAIDs) on preventing and treating Alzheimer disease. The papers cited by the reviewer did show beneficial effects of NSAIDs in the Tg2576 (APP^{Sw} expressed from the PrP promoter), but the data was not well supported by rigorous behavioral studies. In Lim et al., 2001, ibuprofen lowered Il-1 β in piriform and mixed cortices but not in the entorhinal cortex or hippocampus, lowered open field activity in female, but not in male, Tg2576, and lowered A β levels. The behavioral studies done in the Lim et al., 2001 paper did not allow a rigorous assessment of the effect of ibuprofen on AD-related cognition. In Kotilinek et al., they reported an inhibition of A β -induced reduction of EPSPs in rat hippocampal organ slices with COX2 inhibitors. The effect of NSAIDs on spatial memory loss in Tg2576 mice were assessed with the Morris water maze and reported as mean probe scores, which are defined as the mean target occupancy of probes obtained between the 8th and 17th trial. I tried to find more information on this measure, and it seems that it is not one that is usually used in behavioral analyses with the Morris water maze. Assuming that this is a correct measure of spatial memory deficits, the authors failed to exclude factors such as anxiety or motor problems in the Tg2576, which could affect the memory scores. The second behavioral test assessed is the T maze, used to assess spatial working memory. In this test, only Tg2576 were assessed and therefore, it is not possible to affirm that the test accurately distinguishes the AD from normal mice. It is also unclear why the electrophysiology was not done in Tg2576 mice since in addition to the overproduction of A β tested in the organotypic cultures, the neurons would have expressed the mutant APP gene related to familial AD.

The paper by Woodling et al., Brain 2016, which treated the APP/PS1 mouse model of AD with ibuprofen provides compelling evidence for the prevention of cognitive impairment with a three month treatment with ibuprofen given one month before the appearance of deficits on NOR. The prevention of cognitive impairment was independent of A β levels and glial inflammation measured with Iba1. Therefore, VX-765 differs from NSAIDs in that it can rapidly reduce Iba+ microglial activation and GFAP levels in the hippocampus and cortex (Flores et al., Nature communications 2018), and here, we show that the effect is sustained several months after the removal of the drug (Figure 3).

How does VX-765 differ from NSAIDs in terms of anti-inflammatory effect? VX-765 is a selective inhibitor of Caspase-1. Caspase-1 is activated by inflammasomal intracellular receptors that get activated by danger and pathogen molecular patterns (DAMP and PAMP). NSAIDs used in human clinical trials inhibit cyclooxygenases but not the inflammasome-mediated inflammation pathway. Recently fenamate NSAIDs (Daniel et al., Nat Comm 2016) have been able to prevent cognitive impairment in a rat and a mouse AD model, but these have not yet been tested in human clinical trials. We found evidence in the literature of another link between Cox2 and inflammasome. Long coding RNA, lincCox2, located upstream of the mouse Cox2 gene increases expression of Cox2 in Tlr-stimulated bone marrow derived macrophage or dendritic cells. LincCox2 also binds NF-kB p65 and promotes Nlrp3 and ASC expression upon LPS stimulation of bone marrow derived macrophages and the microglial BV2 cell line (Xue et al., Cell Death and Differentiation 2018). As a consequence, Casp1 is activated. Therefore, NSAID inhibition of Cox2, while blocking an important inflammatory pathway, would still allow for Casp1 activation via the NF-kB pathway. In contrast, VX-765 Casp1 inhibitor would block Casp1-mediated inflammation in microglia.

Furthermore, we have shown that Nlrp1 immunostaining increases 20 fold in AD neurons compared to non-cognitively impaired brains and in primary human neuron cultures, Nlrp1

activates Casp1 (Kaushal et al., Cell Death and Differentiation 2015) and Casp1 activates Casp6 (Guo et al., Cell Death and Differentiation 2006). We and others have shown that active Casp6 is highly abundant in AD neurons, co-localizes with Nlrp1, and induces neuritic degeneration by cleaving a number of proteins that are highly relevant in AD pathology. VX-765 Casp1 inhibitor would also block this degenerative path in neurons and would not only act on microglia, as we explained in the discussion..

Why did NSAIDs show beneficial effects in animal models of AD but not in human AD? There is always a risk that animal pre-clinical results in any study will not translate into humans. This can be due to a number of factors, including a poor understanding of the cause of the disease. We think that our work which confirmed the activation of the Nlrp1-Casp1-Casp6 neurodegenerative pathway in human neurons in primary cultures, at a very early stage of AD pathogenesis in human brains, and the link between this pathway and many of the pathologies observed in AD brains, increases our chances of success during clinical studies, but we really will only know once we conduct the clinical trials. At least, it is a new path to test. The current method for finding drug treatments today is based on the identification of a pathogenic pathway in the disease and targeting this pathway in an empirical manner to find treatments. Success in mouse models is an essential step to the eventual performance of clinical trials. Unfortunately, this approach leads to many failures, but it is the normal way to proceed to find new treatments.

In response to the reviewer, we have added a summary of these data in the Discussion's second to last paragraph. We did not discuss everything to keep a focus on the VX-765.

2. What is the expected brain concentration of VX-765 at the dosing regimen used here? Previous publication from this group measured the concentrations in mouse cortex and hippocampus, at 50 mg/kg. This could be added as a comment in the description of the study. "Mice were dosed at 50 mg/kg....expected brain concentrations of xx uM(ref)..."

Our previous paper (Flores et al., 2018) showed brain concentrations of VX-765 and its active metabolite, VRT-043198, in the brain after 50 mg/kg dose of VX-765 via carotid injections. We added this as well as other relevant information at the end of our introduction on page 5.

Knowing the limitations of measuring brain VX-765 concentrations after carotid injections, we decided to expand on VX-765's pharmacokinetic profile and looked at plasma, brain, and CSF concentrations of VX-765 and VRT-043198 after a single IP injection of 50 mg/kg VX-765. The results of this study are presented on pages 5-6, Figure 1, and Tables 1-3. In short, we found that VX-765 can enter the brain at concentrations that are sufficient to inhibit Casp1 after a single injection of 50 mg/kg by IP, the dose used to obtain the effects on cognition.

3. Since washout and continuation of effect after stopping treatment is an important point of this paper, the half-life of VX-765 (and perhaps the active metabolite VRT-043198?) in mouse brains should be reported. If it has been reported elsewhere, please provide reference similar to the expected brain concentration of compound as above.

The half-life of VX-765 has only been reported for in vitro preparations and is reported in the introduction (page 5). To the best of our knowledge, we are not aware of any published reports of VX-765's half-life in vivo. Therefore, we performed the pharmacokinetic study (described above) with the data, including half-life, presented in Tables 1-3 of the result section.

4. Page 5, line 93: Please define NOR at first instance of use.

This has been done.

5. Page 6, lines 103-110: This discussion seems to describe improved performance in NOR testing for VX-765 treated WT mice after 12-week WO as compared to the initial test at 4 months of age. At the same time, the NOR performance decreased in vehicle treated mice between 12-20 months of WO, but did not decrease in VX-765 mice. In Fig 1b, it almost appears as if there is improvement in the WT+VX treated group from 4 to 12 weeks that is then sustained at 20 weeks, even though the performance of the WT+veh group declines over the same period. This should be commented on.

We agree with this observation and had described it in the result section. We had commented on this in the discussion section, lines 319 to 329 (now lines 363-373). We wrote “VX-765’s beneficial effect on WT mice exposed a subtle age-related cognitive decline in normal mice and suggests that VX-765 acts on a pathway underscoring cognitive abilities that is common to aging and AD. VX-765 eliminated age-dependent decline in the NOR discrimination index of WT mice. In the Barnes maze, training performance, and probe day primary latency and errors were significantly improved in the VX-treated WT mice at 12-week WO. A non-significant reduction in primary error and latency was also observed at the 20-week WO. Aging is the largest risk factor in AD, so it is not surprising that many of the hallmarks of aging are also present in AD, only accelerated. ‘Inflammaging’ is used to describe the low-grade inflammation that occurs in aging⁵⁶ and is characterized by 1) a microglial shift towards an immunologically primed state^{57,58}, 2) upregulated expression of inflammasome-associated Nlrp3, Nlrc4, Casp1 and ASC⁵⁹, and 3) increased production of pro-inflammatory cytokines IL-6, TNF- α , IL-1 β , and IL-18^{60,61}. Our data suggest that VX-765 acts to inhibit inflammatory and neurodegenerative pathways common in AD- and age-related cognitive decline.”

We added the sentence “VX-765 eliminated age-dependent decline in the NOR discrimination index of WT mice. In the Barnes maze, training performance, and probe day primary latency and errors were significantly improved in the VX-treated WT mice at 12-week WO. A non-significant reduction in primary error and latency was also observed at the 20-week WO.” in this section to highlight the data to which we were referring. We also slightly modified the rest of the paragraph to increase clarity.

6. Page 7, lines 127-129 and Fig 1e: There was a significant difference in primary latency between the WT+veh and WT+VX groups. Can this be explained?

The WT mice were assessed on Barnes maze at 12- and 20-week WO so at the age of 6 and 8 months, respectively. The VX-treated WT mice showed statistically significant improvement on both primary latency and primary errors measures at the 12-week WO but not at the 20-week WO, although there was a trend towards better performance in VX-treated WT mice. We explained this in the discussion section (page 18) as indicated in comment 5 of this reviewer.

7. Page 15, lines 308-313 & Page 16, line 328: In the discussion section, the authors talk about the potential for VX-765 to treat neurodegeneration, yet provide no evidence for changes in neurodegeneration in the mice. The discussion should be restricted to only claim the compound can reduce inflammatory pathways and improve cognition of these mice. It is possible to hypothesize that this could affect neurodegeneration, but no evidence has been provided that this is true.

Etymologically, neurodegeneration means degeneration of neurons. Degeneration is defined by a process of losing structure or function. Therefore, in the strict sense of the word, conditions

that lead to neuronal dysfunction could be defined by neurodegeneration. In the J20 mice, overexpression of APP^{Sw/Ind} in neurons results in episodic and spatial memory impairment and therefore in neurodegeneration.

As described in the introduction of the paper, we have shown in published studies that inflammasomes are activated in primary human neurons by the overexpression of mutant APP (Swedish or London), and that this leads to Casp1 activation, which then activates Casp6. Casp6 is associated with several pathways of neurodegeneration present in AD pathology. This is why we cannot exclude a potential effect of VX-765 directly in neurons. We conclude that clearly VX-765 decreases microglial inflammation. However, we feel it is important to discuss the possibility of another effect of VX-765 on neurons. This is why we added "VX-765 normalizes Iba1 positive microglia numbers and therefore, we can conclude that the action of VX-765 is, in part, through microglia. However, we cannot exclude the possibility that the beneficial action of VX-765 preventative treatment also occurs in neurons. We previously demonstrated that stress by serum deprivation or overexpression of WT or mutant APP in primary human neuron cultures induces a Nlrp1-Casp1-Casp6 neurodegenerative pathway resulting in the aggregation of several proteins including Tau, ubiquitin, and microtubule-associated protein 2^{15,44}. We hypothesized that Casp1 activation in neurons leads to (1) Casp6-mediated neurodegeneration and increased A β production^{22,45,46} and (2) microglial activation via the release of Il-1 β ²⁰. The fact that 1) Il-1 β levels are extremely low in the hippocampus and cortex of J20 mice (shown here and previously¹⁸) and in human neurons compared to easily detectable levels in activated microglia¹⁵, 2) mutant APP expression in J20 mice is regulated by a predominantly neuronal promoter^{38,47}, and 3) cognitive function is generally a measure of neuronal function, supports a putative role for VX-765 in preventing neuronal degeneration."

REVIEWERS' COMMENTS:

Reviewer #2 (Remarks to the Author):

The authors have addressed my previous concerns well. No further comments.

Reviewer #3 (Remarks to the Author):

In this revised manuscript, the authors adequately addressed my main concern, by providing a detailed rationale in the discussion for why caspase-1 inhibition may work even though NSAIDs have failed in human clinical trials.

In addition, the new PK data are informative and helpful.

However, I still take issue with the use of the word "neurodegeneration." If neurodegeneration were defined strictly as any abnormalities in the structure or function of neurons, then our current APP transgenic mouse models would adequately represent human AD. However, the vast majority of researchers in the AD field consider the lack of neurodegeneration, defined as substantial neuron loss and brain atrophy, to be one of the greatest drawbacks of our current APP transgenic mouse lines. The authors themselves even assert that their studies represent the preclinical or prodromal stages of Alzheimer's disease, and go on to suggest that inhibiting caspase-1 may prevent conversion to full-blown dementia. However, since humans at the preclinical or prodromal stages of disease exhibit little-to-no neurodegeneration, it makes even less sense to extrapolate that any potentially beneficial effects of caspase-1 inhibition might be due to the inhibition of neurodegeneration.

Karen H. Ashe, MD, PhD

Response to Reviewer's comments

We thank the reviewers for their critique. Please find below our response.

REVIEWERS' COMMENTS:

Reviewer #2 (Remarks to the Author):

The authors have addressed my previous concerns well. No further comments.

Thank you.

Reviewer #3 (Remarks to the Author):

In this revised manuscript, the authors adequately addressed my main concern, by providing a detailed rationale in the discussion for why caspase-1 inhibition may work even though NSAIDs have failed in human clinical trials. In addition, the new PK data are informative and helpful.

However, I still take issue with the use of the word "neurodegeneration." If neurodegeneration were defined strictly as any abnormalities in the structure or function of neurons, then our current APP transgenic mouse models would adequately represent human AD. However, the vast majority of researchers in the AD field consider the lack of neurodegeneration, defined as substantial neuron loss and brain atrophy, to be one of the greatest drawbacks of our current APP transgenic mouse lines. The authors themselves even assert that their studies represent the preclinical or prodromal stages of Alzheimer's disease, and go on to suggest that inhibiting caspase-1 may prevent conversion to full-blown dementia. However, since humans at the preclinical or prodromal stages of disease exhibit little-to-no neurodegeneration, it makes even less sense to extrapolate that any potentially beneficial effects of caspase-1 inhibition might be due to the inhibition of neurodegeneration.

Karen H. Ashe, MD, PhD

We understand the concern of the reviewer. I think that the definition of neurodegeneration is the issue here. According to the Meriam-Webster Medical Dictionary: neurodegeneration is defined as relating to or marked by degeneration of nervous tissue. Degeneration is defined as : (a) progressive deterioration of physical characters from a level representing the norm of earlier generations or forms or (b) deterioration of a tissue or an organ in which its function is diminished or its structure is impaired. Therefore, according to this definition, we can correctly assume that what we observe is neurodegeneration.

Our studies have shown that Casp6 impairs synaptic function and induces neurodegeneration. In the introduction, we have added "Transfection of human neurons with mutant or wild type APP induces Casp6-dependent, but A β -independent, neuritic degeneration¹. Active Casp6 in the hippocampal Cornu Ammonis 1 (CA1) is sufficient to induce age-dependent cognitive deficits in mice². Theta-burst long term potentiation cannot be initiated in mice acute brain slice hippocampal CA1 neurons expressing active Casp6³. Furthermore, injection of active Casp6 in wild type mice CA1 neurons impairs synaptic transmission and induces neurodegeneration⁴" to support the use of the term degeneration. In the J20 mouse model, the fact that we confirmed the loss of synaptophysin and cognitive function, as had shown Dr Mucke, and that this was reversed by treatment with VX-765 also indicates neurodegeneration⁵. Furthermore, in this paper, the involvement of Casp1 was shown by generating J20 on a Casp1 null background, and this also corrected the functional defect.

In the Alzheimer disease field of research, neurodegeneration is often associated with neuronal cell death. It is also true that most scientists believe that atrophy represents neuronal cell death. Most neuroimagers will indicate that a loss of brain volume in Alzheimer disease is explained by a loss of neurons. This is a definition that can be found on certain web sites (for example www.healthline.com, Wikipedia and others). However, there is no scientific data to support this assumption. In fact, there is evidence to the contrary. In published research where definite stereological cell counts, mostly neurons, has been done in cognitively ascertained human brains, neuronal loss is seen mostly in late stages of Alzheimer disease (demented cases or stage V)^{6,7}, whereas hippocampal atrophy occurs in the early stages of the disease and is now considered as an early marker of the disease. Only one paper indicates a loss of neurons in brains from mild cognitively impaired individuals and this loss of about 50% neurons is limited to the layer II of the entorhinal cortex⁸. The study was done on only 7 different human brains. A later paper from this group indicates loss of neurons in the superior temporal gyrus in the brain of advanced cases of Alzheimer's disease⁹. Considering all of the studies that have been published based on stereological counting, it is safe to say that neuronal loss is fairly limited both in number and in regions in the early phases of the disease.

References

- 1 Sivananthan, S. N., Lee, A. W., Goodyer, C. G. & LeBlanc, A. C. Familial amyloid precursor protein mutants cause caspase-6-dependent but amyloid beta-peptide-independent neuronal degeneration in primary human neuron cultures. *Cell Death Dis* **1**, e100 (2010).
- 2 LeBlanc, A. C. *et al.* Caspase-6 activity in the CA1 region of the hippocampus induces age-dependent memory impairment. *Cell Death Differ* **21**, 696-706 (2014).
- 3 Zhou, L. *et al.* Methylene blue inhibits Caspase-6 activity, and reverses Caspase-6-induced cognitive impairment and neuroinflammation in aged mice. *Acta Neuropathol Commun* **7**, 210 (2019).
- 4 Noel, A., Zhou, L., Foveau, B., Sjostrom, P. J. & LeBlanc, A. C. Differential susceptibility of striatal, hippocampal and cortical neurons to Caspase-6. *Cell Death Differ* **25**, 1319-1335 (2018).
- 5 Flores, J. *et al.* Caspase-1 inhibition alleviates cognitive impairment and neuropathology in an Alzheimer's disease mouse model. *Nat Commun* **9**, 3916 (2018).
- 6 Rossler, M., Zarski, R., Bohl, J. & Ohm, T. G. Stage-dependent and sector-specific neuronal loss in hippocampus during Alzheimer's disease. *Acta Neuropathol* **103**, 363-369 (2002).
- 7 West, M. J., Coleman, P. D., Flood, D. G. & Troncoso, J. C. Differences in the pattern of hippocampal neuronal loss in normal ageing and Alzheimer's disease. *Lancet* **344**, 769-772 (1994).
- 8 Gomez-Isla, T. *et al.* Profound loss of layer II entorhinal cortex neurons occurs in very mild Alzheimer's disease. *J Neurosci* **16**, 4491-4500 (1996).
- 9 Gomez-Isla, T. *et al.* Neuronal loss correlates with but exceeds neurofibrillary tangles in Alzheimer's disease. *Ann Neurol* **41**, 17-24 (1997).